# Effects of Thermodynamics, Dynamics and Aerosols on Cirrus Clouds Based on In Situ Observations and NCAR CAM6 Model

Ryan Patnaude[1], Minghui Diao[1], Xiaohong Liu[2], Suqian Chu[3]

[1]Department of Meteorology and Climate Science, San Jose State University, San Jose, 95192, USA
[2]Department of Atmospheric Sciences, Texas A&M University, College Station, 77843, USA,
[3]Department of Atmospheric Science, University of Wyoming, Laramie, 82071, USA

*Correspondence to:* Minghui Diao (minghui.diao@sjsu.edu)

**Abstract.** Cirrus cloud radiative effects are largely affected by ice microphysical properties, including ice water content (IWC), ice crystal number concentration (Ni) and mean diameter (Di). These characteristics vary significantly due to thermodynamic, dynamical and aerosol conditions. In this work, a global-scale observation dataset is used to examine regional variations of cirrus cloud microphysical properties, as well as several key controlling factors, i.e., temperature, relative humidity with respect to ice (RHi), vertical velocity (w), and aerosol number concentrations (Na). Results are compared with simulations from the National Center for Atmospheric Research (NCAR) Community Atmosphere Model version 6 (CAM6). Observed and simulated ice mass and number concentrations are constrained to $\geq 62.5$ µm to reduce potential uncertainty from shattered ice in data collection. The differences between simulations and observations are found to vary with latitude and temperature. Comparing with averaged observations at ~100 km horizontal scale, simulations are found to underestimate (overestimate) IWC by a factor of 3–10 in the Northern (Southern) Hemisphere. Simulated Ni is overestimated in most regions except Northern Hemisphere midlatitude. Simulated Di is underestimated by a factor of 2, especially for warmer conditions (-50ºC to -40ºC), possibly due to misrepresenting ice particle growth/sedimentation. For RHi effects, the frequency and magnitude of ice supersaturation is underestimated in simulations for clear-sky conditions. The simulated IWC and Ni show bimodal distributions with maximum values at 100% and 80% RHi, differing from the unimodal distributions that peak at 100% in the observations. For w effects, both observations and simulations show variances of w ($\sigma_w$) decreasing from tropics to polar regions, but simulations show much higher $\sigma_w$ for in-cloud condition than clear-sky condition. Compared with observations, simulations show weaker aerosol indirect effects with smaller increase of IWC and Di at higher Na. These findings provide an observation-based guideline for improving simulated ice microphysical properties and their relationships with key controlling factors at various geographical locations.

## 1 Introduction

Cirrus clouds represent one of the most ubiquitous cloud types with an estimated global coverage of approximately 20% to 40% (Mace and Wrenn, 2013; Sassen et al., 2008). According to the fifth assessment of the United Nations Intergovernmental Panel on Climate Change (IPCC) report (Boucher et al., 2013), the largest uncertainty in estimating future

climate change stems from clouds and aerosols. Unlike most other cloud types, cirrus clouds may produce a net positive or negative radiative forcing depending on their microphysical properties (Stephens and Webster, 1981; Zhang et al., 1999), which are affected by meteorological conditions and aerosol distributions. Tan et al. (2016) showed that the radiative effects of misrepresenting the prerequisite condition of cirrus clouds – ice supersaturation (ISS, where relative humidity with respect to ice (RHi) > 100%) – can lead to an average bias of +2.49 W/m$^2$ at the top of the atmosphere. Other modelling studies found large differences in the net cloud radiative forcing depending on the fraction of activated ice nucleating particles (INPs) and the nucleation mechanisms (i.e., homogeneous and heterogeneous nucleation) through which the clouds form (Liu et al., 2012; Storelvmo and Herger, 2014). The large uncertainties in cirrus cloud radiative forcing illustrate the need for further study on cirrus cloud microphysical properties as well as their controlling factors in various geographical locations.

Ideally, a comprehensive quantification of cirrus cloud microphysical properties globally based on high-resolution, in situ observations would mitigate many uncertainties. However, challenges remain in field measurements to achieve such spatial coverage. Previously, efforts have been made to understand cirrus cloud properties based on their geographical locations. Diao et al. (2014b) performed a hemispheric comparison of in situ cirrus evolution and found little difference in the clear-sky ISS frequency as well as the proportion of each evolution phase between the Northern and Southern Hemispheres (NH and SH, respectively). In situ observations of tropical, midlatitude, and polar cirrus clouds have shown that IWC can vary orders of magnitude depending on the geographical locations (Heymsfield, 1977; Heymsfield et al., 2005, 2017; Mcfarquhar and Heymsfield, 1997; Schiller et al., 2008). Wolf et al. (2018) used balloon-based in situ observations to analyze microphysical properties of Arctic ice clouds and found differences in particle size distributions (PSDs) depending on the cloud origin. Krämer et al. (2016, 2020) developed a cirrus cloud climatology, focusing on tropical and midlatitude cirrus clouds, and showed that cloud thickness is larger at lower altitudes, and thus producing a more negative radiative forcing. Moving from north to south using lidar-based observations from two research cruises starting from Leipzig, Germany, one to Punta Arenas, Chile and the other one to Stellenbosch, South Africa, Kanitz et al. (2011) observed a decrease in the efficiency of heterogeneous nucleation in the SH, which could be a result of fewer INPs. This hemispheric difference in aerosol indirect effects is consistent with significantly higher aerosol number concentrations in the NH (Minikin et al., 2003). Using satellite observations from the Cloud-Aerosol Lidar and Infrared Pathfinder Satellite Observation (CALIPSO), Mitchell et al. (2018) showed the dependence of ice particle effective diameter on temperature, latitude, season and topography. Thorsen et al. (2013) used CALIPSO data to examine cloud fraction of tropical cirrus clouds and showed dependence on altitude and diurnal cycle. Tseng and Fu (2017) used CALIPSO and Constellation Observing System for Meteorology, Ionosphere, and Climate (COSMIC) data and found that the tropical cold point tropopause temperature is a controlling factor of cirrus cloud fraction in the tropical tropopause layer.

Regional and hemispheric variations of cirrus microphysical properties are produced by various controlling factors, such as thermodynamics (i.e., temperature and RHi), dynamics (e.g., vertical velocity) and aerosols (e.g., number concentration and composition). The effects of temperature have been extensively studied from in situ observations (Heymsfield et al., 2017; Luebke et al., 2013, 2016; Schiller et al., 2008), showing an increase of IWC towards warmer temperatures. A number of

studies focused on distributions of RHi have found that in-cloud RHi occurs most frequently at or near 100% (Jensen et al., 2001; Krämer et al., 2009). Another study by Diao et al. (2017) found that using different RHi thresholds (e.g., 108% to 130%) for ice nucleation in simulations can influence IWC and ice crystal number concentrations (Ni) in convective cirrus. In addition, the spatial scales of ice supersaturated regions can vary from the micro- to mesoscales, largely depending on the spatial variability of water vapor (Diao et al., 2014a). The distributions of vertical velocity have been investigated in

different types of cirrus clouds, such as in ridge-crest cirrus, frontal cirrus and anvil cirrus (Muhlbauer et al., 2014a, 2014b). Stronger updrafts are found to be associated with higher occurrence frequency of ISS inside anvil and convective cirrus (D'Alessandro et al., 2017). Regarding the effects of aerosols, Cziczo et al. (2013) and Cziczo and Froyd (2014) investigated ice crystal residuals from in situ observations and discovered that the majority of midlatitude cirrus clouds form via heterogeneous nucleation on mineral dust and metallic particles. Anthropogenic aerosols, such as secondary organic

aerosols, were found to be less effective INPs compared with mineral dust (Prenni et al., 2009). Based on remote sensing data, Zhao et al. (2018, 2019) showed that the correlations between ice crystal sizes and aerosol optical depth can be either positive or negative depending on the meteorological conditions in convective clouds. Chylek et al. (2006) showed an increase in ice crystal size during the more polluted winter months compared with cleaner summer months over the eastern Indian Ocean, which the authors speculate to be due to heterogeneous nucleation occurring at lower ice supersaturation

compared with homogeneous nucleation, therefore reducing the ambient ice supersaturation magnitude and making homogeneous nucleation a more difficult pathway. Using a global-scale dataset of multiple flight campaigns, Patnaude and Diao (2020) isolated individual effects on cirrus clouds from temperature, RHi, vertical velocity (w) and aerosol number concentrations (Na). They found that when Na is 3 – 10 times higher than average conditions, it shows strong positive correlations with cirrus microphysical properties such as IWC, Ni and number-weighted mean diameter (Di). These aerosol

indirect effects are also susceptible to whether or not thermodynamic and dynamical conditions are controlled, demonstrating the importance of conducting a comprehensive analysis of various key controlling factors altogether.

     More recently, in situ observations have been used to evaluate and improve cirrus cloud parameterizations in global climate models (GCMs). Two types of simulations have been frequently used for model evaluation, i.e., free-running (Eidhammer et al., 2014, 2017; Wang and Penner, 2010; Zhang et al., 2013) and nudged (D'Alessandro et al., 2019; Kooperman et al., 2012;

Wu et al., 2017) simulations. For free-running simulations, a comparison on statistical distributions of ice microphysical properties is often used for model validation (e.g., Penner et al., 2009). The nudged simulation would nudge certain meteorological conditions towards reanalysis data, such as horizontal wind and temperature (e.g., D'Alessandro et al., 2019; Wu et al., 2017). These nudged simulations can also be output to similar location and time as those of the aircraft observations. Given the importance and limited understanding of how aerosols interact with cirrus clouds, much attention

has been dedicated to the parameterization of aerosol indirect effects (Kärcher and Lohmann, 2002, 2003; Kuebbeler et al., 2014; Wang et al., 2014a). Shi et al. (2015) added the effects of pre-existing ice into the Community Atmosphere Model Version 5 (CAM5) and found a decrease in Ni with increasing aerosol concentration due to the reduction of homogeneous

nucleation frequency. Other studies also investigated the effect of updraft velocity on simulated Ni and aerosol indirect effects (Zhou et al., 2016; Penner et al., 2018).

This study aims to bridge the knowledge gap on how cirrus clouds vary depending on geographical locations and environmental conditions by using a comprehensive in situ observation dataset that includes seven U.S. National Science Foundation (NSF) flight campaigns. Observations were collected onboard the NSF/National Center for Atmospheric Research (NCAR) Gulfstream-V (G-V) research aircraft. Descriptions of the seven flight campaigns, instrumentations, model configurations of the NCAR Community Atmosphere Model version 6 (CAM6) are provided in Section 2. Both

observations and simulations are used to examine the regional variations in the statistical distributions of cirrus microphysical properties, including IWC, Ni and Di (Section 3). Impacts of several key controlling factors, i.e., temperature, RHi, w and Na, are examined in Section 4. Discussions on observation-based findings and model evaluation results are included in Section 5.

## 2 Data and Methods

### 2.1 In situ observations and instrumentations

In this study, in situ airborne observations at 1 Hz are provided by instruments onboard the NSF High-Performance Instrumented Airborne Platform for Environmental Research (HIAPER) G-V research aircraft. A comprehensive global dataset is compiled based on seven major flight campaigns funded by the NSF, including START08 (Pan et al., 2010), HIPPO deployments 2–5 (Wofsy et al., 2011), PREDICT (Montgomery et al., 2012), TORERO (Volkamer et al., 2015),

DC3 (Barth et al., 2015), CONTRAST (Pan et al., 2017), and ORCAS (Stephens et al., 2018). Table 1 provides a detailed summary of the seven flight campaigns, including location, duration of flights, total flight hours of all temperatures, and flight hours for in-cloud and clear-sky conditions at temperatures ≤ -40ºC only. Maps comparing the flight tracks of in situ observations and the collocated CAM6 nudged simulations (hereafter named "CAM6-nudg" data) are shown in Figure 1.

For this study, ice particle measurements are provided by the Fast 2-Dimensional Cloud particle imaging probe (Fast-2DC)

with a 64-diode laser array for a range of 25 μm – 1600 μm. Larger particles can be reconstructed up to 3200 μm. The mass-Dimensional relationship of Brown and Francis (1995) is used to calculate IWC for the Fast-2DC probe, which was previously used in other studies of the Fast-2DC probe onboard the NSF G-V aircraft (Diao et al., 2014a, 2014b, 2015). Number-weighted mean diameter (Di) is calculated by summing up the size of particles in each bin using the bin center, and then dividing it by the total number of particles. In order to mitigate the shattering effect, particles with diameters < 62.5 μm

(i.e., first two bins) are excluded in the Fast-2DC measurements when calculating IWC, Ni and Di. The Rosemount temperature probe was used for temperature measurements, which has an accuracy and precision of ~ ±0.3 K and 0.01 K, respectively. All analyses are restricted to temperatures ≤ -40°C, in order to exclude the presence of supercooled liquid droplets in this study. Laboratory calibrated and quality-controlled water vapor data were collected using the Vertical Cavity

Surface Emitting Laser (VCSEL) hygrometer (Zondlo et al., 2010), with an accuracy of ~6% and precision of ≤ 1%. Both temperature and water vapor are used at 1-Hz resolution for this analysis. Aerosol measurements were collected from the Ultra-High Sensitivity Aerosol Spectrometer (UHSAS), which uses 100 logarithmically spaced bins ranging from 0.06 – 1 µm. RHi is calculated using saturation vapor pressure with respect to ice from Murphy and Koop (2005). The combined RHi uncertainties from the measurements of temperature and water vapor range from 6.9% at -40ºC to 7.8% at -78ºC. Measurements are separated by cloud condition where in-cloud condition is defined by the presence of at least one ice crystal from the Fast 2-DC probe ($Ni > 0$ $L^{-1}$). The same in-cloud definition has been used by several previous studies (D'Alessandro et al., 2017; Diao et al., 2014a, 2014b, 2015, 2017; Tan et al., 2016), and all other samples are defined as clear sky. For regional variation analysis, data are binned by six latitudinal regions in the two hemispheres, that is, NH polar (60ºN – 90ºN), SH polar (60ºS – 90ºS), NH midlatitude (30ºN – 60ºN), SH midlatitude (30ºS – 60ºS), NH tropics (0º – 30ºN), and SH tropics (0º – 30ºS). The majority of observations in the SH midlatitude and tropical regions are located over the oceans, while the observations of NH midlatitude and polar regions are predominantly over land.

The vertical profiles of observed in-cloud temperature, clear-sky potential temperature ($\Theta$), and their correlations are shown in Figure 2. The observations sampled temperatures from -78ºC to -40ºC and altitudes from 5 – 15 km, while a previous study of Krämer et al. (2020) sampled -91ºC to -30ºC and 5 – 19 km (their Figure 2). The lowest temperatures are found in the tropical regions and at the highest altitudes, whereas polar regions show more observations at lower altitudes that satisfy temperature ≤ -40ºC. Distributions of cirrus cloud properties (i.e., IWC, Ni, Di), in-cloud and clear-sky RHi, and clear-sky water vapor mixing ratio for the observation dataset are shown in Figure 3. Di increases with decreasing altitudes, IWC slightly increases with decreasing altitudes, and Ni is almost independent of altitudes. Clear-sky RHi and water vapor mixing ratio both increase with decreasing altitudes, while in-cloud RHi is centered around 100% and shows smaller dependency on altitudes. Compared with Figure 3 in Krämer et al. (2020), 48% of their ice particle samples have Di < 40 µm, which is below the size cut-off used in this study. The higher Di in this study also leads to lower range of Ni (0.01 – 1000 $L^{-1}$) and higher range of IWC ($10^{-5}$ – 10 g m$^{-3}$) compared with that previous study (i.e., Ni from 0.1 – $10^5$ $L^{-1}$ and IWC from $10^{-7}$ – 1 g m$^{-3}$), representing the sampling bias towards larger particles in this study. The relationships of IWC with respect to meteorological conditions (i.e., temperature and RHi) and other microphysical properties (i.e., Ni and Di) are shown in supplementary Figure S1. The distributions of IWC samples are relatively uniform at various temperature and RHi, while more IWC samples are correlated with Di between 100 and 300 µm.

## 2.2 Climate model description and experiment design

This study uses model simulations based on the NCAR CAM6 model. Compared with its previous version – the CAM5 model, CAM6 implemented a new scheme, the Clouds Layers Unified by Binomials (CLUBB) for representations of boundary layer turbulence, shallow convection and cloud macrophysics (Bogenschutz et al., 2013). CLUBB is a higher-order turbulence closure scheme that calculates prognostic higher-moments based on joint probability density function (PDFs) for vertical velocity, temperature, and moisture (Golaz et al., 2002). An improved bulk two-moment cloud microphysics scheme

has been implemented (Gettelman and Morrison, 2015) that replaces diagnostic treatment of rain and snow with prognostic treatment of all hydrometeors (i.e., rain and snow). This is coupled with a 4-mode aerosol model (MAM4) (Liu et al., 2016) for simulations of aerosols and aerosol-cloud interactions. It allows ice crystals to form via homogeneous freezing of sulfate

aerosols and heterogeneous nucleation of dust particles (Liu et al., 2007; Liu and Penner, 2005). The model uses Wang et al. (2014b) for ice nucleation, which implemented and improved Hoose et al. (2010) by considering the probability density function of contact angles for the classical nucleation theory. The model also uses Shi et al. (2015) for modifications of pre-existing ice. Finally, the deep convection scheme (Zhang and McFarlane, 1995) has been tuned to include sensitivity to convection inhibition.

Results from in situ observations are compared with two types of CAM6 simulations, nudged and free-running simulations. Simulations are based on a finite-volume dynamical core (Lin, 2004) with a horizontal resolution of 0.9°×1.25° and 32 vertical levels. All simulations are conducted using prescribed sea-surface temperature and present-day aerosol emissions and include a 6-month spin-up time. CAM6 nudged simulations are nudged spatially and temporally with meteorological data (i.e., 2-D horizontal wind and temperature) from the Modern-Era Retrospective Analysis for Research and Applications

version 2 (MERRA2) (Gelaro et al., 2017), and collocated with aircraft flight tracks in space and time. A nudged simulation was conducted for each campaign independently and was combined into one data set to compare with observations. One free-running simulation was conducted for the duration of all flight campaigns from July 2008 to February 2016. To reduce the size of model output when comparing with observations, a total of 24 instantaneous output from the free-running simulation are combined into one data set ("CAM6-free" hereafter), which includes 00 and 12 UTC for the first day of each

month in 2010. Additional sensitivity tests on different model output from the free-running simulation show very minor differences in the statistical distributions of cirrus microphysical properties and the correlations with their controlling factors when selecting different years, seasons, and days in a month.

In order to examine observations and simulations on more comparable scales, a running average of 430 seconds was calculated for meteorological parameters (i.e., temperature and RHi) and microphysical properties (i.e., IWC, Ni and Di),

which translates to ~100 km horizontal scales since the mean true air speed below -40°C for all campaigns was 230 m/s (supplementary Figure S2). When applying the running average, both in-cloud and clear-sky conditions (i.e., where Ni and IWC values are zero) are included in the averages. Grid-mean quantities from model output are used in comparisons with observations, including "IWC", "NUMICE", "QSNOW" and "NSNOW", which are mass and number concentrations of ice particles and snow, respectively. Another type of comparison between 1-Hz observations and in-cloud quantities from model

output is shown in the supplementary material. Both methods have been previously used in model evaluation, such as D'Alessandro et al. (2019) which compared 200-s averaged aircraft observations with simulated grid-mean quantities, and Righi et al. (2020) which compared 1-Hz aircraft observations with simulated in-cloud quantities.

A summary of the ranges of meteorological conditions and ice microphysical properties for in situ observations and simulations is shown in Table 2. Simulated RHi is calculated from simulated specific humidity and temperature, and the

calculation of saturation vapor pressure with respect ice is based on the equation from Murphy and Koop (2005). Simulated

ice and snow are restricted to ≥ 62.5 μm based on the size cut-off of the Fast-2DC probe by applying methods from Eidhammer et al. (2014). Based on their equations 1 to 5, we followed their assumption that the shape parameter μ equals 0 when calculating the slope parameter λ. Mass and number concentrations of ice and snow are further calculated based on integrals of incomplete gamma functions from 62.5 μm to infinity. The simulated values of IWC, Ni and Di are calculated based on the combined ice and snow population after applying the size restriction. In-cloud conditions in simulations are defined by concurring conditions of IWC > $10^{-7}$ g m$^{-3}$ and Ni > $10^{-4}$ L$^{-1}$ based on size-restricted grid-mean quantities. These thresholds are the lower limits from observations after calculating the 430-s averages. Note that due to the ice crystal size constraint, some thin cirrus may not be detected. In addition, analysis of simulated cirrus clouds is restricted to similar pressure ranges as those measured in the seven campaigns. An additional constraint on cloud fraction > $10^{-5}$ was applied to both nudged and free-running simulations to exclude extremely low values.

To visualize the impact of the size truncation on simulated data, we employed methods similar to Gettelman et al. (2020) and reconstructed the simulated particle size distributions for snow and ice in Figure 4 a, using gamma functions from Morrison and Gettelman (2008). Compared with the observations, the number density for combined ice and snow is overestimated for smaller particles (< 400 μm) and underestimated for larger particles (> 1000 μm). After applying size restriction, the PDF of simulated Ni and IWC show increasing probability of small Ni and decreasing probability of small IWC due to the removal of small particles (Figure 4 b and c).

Finally, simulated aerosols number concentrations are further categorized by diameters > 500 nm and > 100 nm (i.e., Na$_{500}$ and Na$_{100}$, respectively), by summing the size-restricted concentrations of the Aitken, accumulation and coarse aerosol modes. Previously, field experiments found that Na$_{500}$ correlates well with INP number concentrations (DeMott et al., 2010). Even though that correlation was only determined based on observations warmer than -36ºC, the separation of Na$_{500}$ and Na$_{100}$ can help to examine the effects of larger and smaller aerosols in this work.

## 3 Regional variations of cirrus cloud characteristics

### 3.1 Cirrus cloud microphysical properties with respect to temperature

Three cirrus cloud microphysical properties, IWC, Ni and Di are examined in relation to temperature at six latitudinal regions (Figure 5), the standard deviations of the IWC, Ni and Di in each temperature bin are shown in supplementary Figure S3. The 1-Hz observations of IWC and Ni in the NH indicate clear latitudinal differences with the highest values occurring in the midlatitudes, followed by tropics, then polar regions for temperatures between -40ºC and -60ºC, while for colder temperatures the NH tropical region shows the highest IWC. In the SH, the highest IWC and Ni occur in the tropics, followed by the polar regions and midlatitudes. Comparing the two hemispheres, IWC and Ni show significant reductions by ~1 order of magnitude from NH midlatitude to SH midlatitude (Figure 5 b, e). These hemispheric differences in midlatitudes may be due to airmass differences between NH (more continental) and SH (more oceanic) and/or more anthropogenic

emissions in the NH. The IWC, Ni and Di are relatively similar between NH and SH tropical regions, while IWC and Di are higher in the SH polar region than NH polar regions.

The simulations are further compared with averaged observations at a similar horizontal scale of ~100 km. After applying 430-s running averages for observations, the average IWC and Ni values decrease by 0.5 – 1.5 orders of magnitude compared with 1-Hz observations depending on temperature and geographical region. Hemispheric differences are mostly consistent between 1-s and 430-s averaged observations except for polar regions. CAM6-nudg data show similar trend of average IWC, Ni and Di with respect to temperature as seen in observations, that is, the average IWC increases with increasing temperature consistent with previous observational studies (Krämer et al., 2016; Luebke et al., 2013; Schiller et al., 2008), average Ni shows no clear trend with temperature, and average Di increases with increasing temperature. Differing from observations, CAM6 produces the highest IWC and Ni in the tropical regions, followed by midlatitudes then polar regions for both hemispheres. The simulated IWC, Ni and Di also show smaller differences between hemispheres and latitudes. The CAM6-nudg data underestimate and overestimate IWC in the NH and SH by 0.5 – 1 orders of magnitude, respectively, with the largest discrepancies in the midlatitudes. The simulations overestimate Ni in the tropics and polar regions in both hemispheres by 0.5 – 1 orders of magnitude, and overestimate Ni in the southern hemispheric midlatitude by 1 – 2 orders of magnitude. The simulated Di is about half of the observed values in most regions except polar regions. This result indicates "too many" and "too small" simulated ice in most regions. The low bias of simulated Di indicates possible misrepresentation of ice particle growth and sedimentation in the model parameterization.

A sensitivity test is conducted by comparing 1-Hz observations with in-cloud quantities from model output (supplementary Figure S4). Larger differences are seen between simulated and observed IWC and Ni in Figure S4 compared with Figure 5. The directions (i.e., positive or negative) of model biases of IWC, Ni and Di are generally consistent in both comparisons.

A previous study by Righi et al. (2020) evaluated the ice microphysical properties in EMAC-MADE3 aerosol–climate model (i.e., ECHAM/MESSy Atmospheric Chemistry-Modal Aerosol Dynamics model for Europe adapted for global applications, 3rd generation) by comparing in-cloud quantities from model output with 1-Hz in situ observations of multiple aircraft field campaigns from 75ºN to 25ºS (Krämer et al., 2009, 2016, 2020). Although that study included more smaller ice particles (3 – 1280 µm) compared with this study, they still showed low biases of simulated Di at 190 – 243 K, low biases of simulated IWC at 205 – 235 K, as well as high biases of simulated Ni above 225 K, which are generally in the same direction as the biases we found in CAM6 model. Note that Righi et al. (2020) implemented different cloud microphysics parameterizations compared with the CAM6 model, including a two-moment cloud microphysics scheme of Kuebbeler et al. (2014) and the ice nucleation parameterization for cirrus clouds (T < 238.15 K) from Kärcher et al. (2006) which account for both homogeneous and heterogeneous nucleation and the competition between the two mechanisms. More future intercomparison studies of these models are warranted to examine the reasons behind the similar biases.

### 3.2 RHi and σ_w distributions for in-cloud and clear-sky conditions

Regional distributions of RHi for clear-sky and in-cloud conditions are shown for 1-Hz observations (Figure 6), 430-s

averaged observations (Figure 7) and simulations (Figure 8). The 1-Hz observations show RHi magnitudes ranging from <
5% up to ~180% in both clear-sky and in-cloud conditions, and mostly locate below the homogeneous freezing line except
for the NH tropical region. A few samples exceed liquid saturation line but are within the measurement uncertainties of RHi.
This result agrees with the RHi distributions based on previous midlatitudinal observations (Cziczo et al., 2013). Differing
from 1-Hz observations, 430-s averaged observations show much lower RHi magnitudes for both clear-sky and in-cloud

conditions, ranging from < 5% to 120% – 140%. For clear-sky conditions, the majority of the observed and simulated RHi
values are below 100%, while the CAM6-nudg data show fewer RHi exceeding ice saturation. For in-cloud conditions, both
1-Hz observations and simulations show that RHi frequently occur within ~20% of ice saturation, consistent with previous
observation and modeling studies (Diao et al., 2014a, 2017; D'Alessandro et al., 2017, 2019; Krämer et al., 2009), while
almost no simulated RHi data exceed the homogeneous freezing threshold. The higher RHi observed in the NH tropical

region was also observed by Krämer et al. (2009). Such feature can be explained by the competition between higher updrafts
seen in the tropics and the depletion of water vapor from newly nucleated ice particles as discussed in Kärcher and Lohmann
(2002). For the polar regions, in-cloud RHi is skewed towards ISS in both observations and simulations, indicating less
effective water vapor depletion likely due to lower $N_i$ values (Figure 5 e). Note that the simulation samples in the tropical
regions show peak frequencies at certain temperatures due to larger bin sizes of pressure levels in the lower latitudes.

Regional distributions of the variance of w ($\sigma_w$) for 1-Hz observations at 40-s and 430-s scales and CAM6 nudged
simulations are shown in Figures 9, 10 and 11, respectively. $\sigma_w$ in the observations is calculated as the variance of w within
each 40 and 430 seconds of data, which correspond to a horizontal scale of ~10 and 100 km, respectively. The $\sigma_w$ in
simulations is based on the "wsub" variable, which is calculated from the square root of turbulent kinetic energy (TKE)
(Gettelman et al., 2010). Observed $\sigma_w$ shows the highest values in the tropical and midlatitude regions reaching up to ~3 m/s,

while the polar regions show updrafts up to ~1 m/s. A similar decreasing trend of maximum $\sigma_w$ is seen in the simulations
from the lower to higher latitudes. The observations show similar $\sigma_w$ maximum values between clear-sky and in-cloud
conditions, while the simulations show much higher maximum $\sigma_w$ for in-cloud conditions in the tropics (1 m/s), midlatitude
(1 m/s) and polar regions (0.5 m/s), compared with those values in clear sky (i.e., 0.5, 0.25 and 0.1 m/s, respectively). This
result suggests that the model has a stronger dependence on higher $\sigma_w$ for cirrus cloud formation compared with

observations. We further examine the potential impact of convection in simulations and observations. Supplementary Figure
S5 shows the locations where w > 1 m/s is seen in the observations as well as where wsub > 0.5 m/s is seen in the CAM6-
nudg data for in-cloud conditions. Since wsub in CAM6 is based on the turbulent scheme, higher wsub values indicate that
the convection scheme may be active and produce detrained ice in convective outflows. The majority of observed and
simulated in-cloud samples do not appear to have high w or wsub, indicating that detrained ice from the convection is

unlikely a significant contribution. More future investigation is needed to track cirrus cloud origins and quantify impacts from convection.

## 4 Individual impacts of key controlling factors on cirrus clouds

### 4.1 Probability density functions of temperature, RHi and $\sigma_w$

PDFs of temperature, RHi and $\sigma_w$ are shown in Figure 12. The PDFs are normalized by the total number of samples of both
clear-sky and in-cloud conditions. The observations are located mostly around -68ºC to -40ºC, and the simulations show similar temperature distributions. For the PDFs of RHi, the observations and simulations all show peak position around 100% for in-cloud condition. However, a secondary peak is shown in simulations at 80% RHi, which is likely due to the parameter of $RHi_{min}$ for ice cloud fraction calculation being set at 80% for representing variance of humidity in a grid box (more details on $RHi_{min}$ are described in Gettelman et al. (2010)). In addition, the maximum RHi for in-cloud conditions are
170%, 154%, 160% and 257% for 1-Hz observations, 430-s averaged observations, CAM6-nudg and CAM6-free, respectively. The maximum RHi for clear-sky conditions are 175%, 135%, 108%, and 181%, respectively. The CAM6-free data show higher maximum RHi values than CAM6-nudg data, likely due to additional data from tropical regions at temperatures below -70ºC (Figure 12 j). When using a lower size cut-off (1 μm) of ice particles for the simulation data, the number of in-cloud samples increases by 4% (supplementary Figure S6). However, negligible differences are seen in the
PDFs of temperature, RHi and $\sigma_w$ for the two simulations between Figures 12 and S6.

PDFs of $\sigma_w$ show consistent results to Figures 9 – 11, with simulations showing much higher maximum $\sigma_w$ for in-cloud conditions than clear-sky conditions, while observations show similar maximum $\sigma_w$ in both conditions. The lower maximum values of $\sigma_w$ in simulations are most likely a result of model missing representations of gravity waves from topography, fronts, and convection, and only including $\sigma_w$ from turbulence.

### 310 4.2 Effects of RHi and $\sigma_w$ on ice microphysics

The relationships between ice microphysical properties and RHi are examined in Figure 13. For the 1-Hz observations, the maximum IWC and Ni occur slightly above ice saturation at 110% RHi, while the maximum Di occur at 130% RHi. The average IWC and Ni increase 1.5 orders of magnitude from 40% to 110% RHi, and decrease 0.5 order of magnitude (i.e., a factor of 3) from 110% to 130% RHi. The maximum IWC and Ni do not occur at the highest RHi most likely due to the
consumption of water vapor by ice deposition. High Di values at lower RHi (~30%) are likely a result of sedimenting large ice crystals, which has been previously observed by Diao et al. (2013) when investigating the evolutionary phases of cirrus clouds. For 430-s averaged observations, the peak IWC, Ni and Di occur at 100%, 100% and 115% RHi. The maximum IWC and Ni values are nearly the same between 1-s and 430-s averaged observations (i.e., 0.04 g/m$^3$ and 10 #/L, respectively) near saturation, while the 430-s averaged observations show lower minimum IWC and Ni at very low RHi (<
10%), which are 1.5 orders of magnitude lower than 1-Hz observations. This feature is due to in-cloud segments being

longer around saturation compared with subsaturated conditions as shown in Diao et al. (2013), which means that less clear-sky conditions are being included in the 430-s averages around saturation and therefore show little reduction of the IWC and Ni due to spatial averaging.

In contrast to observations, both CAM6-nudg and CAM6-free simulations show bimodal distributions of IWC and Ni with the primary peak at 100% RHi and the secondary peak at 80% RHi. The secondary peak at RHi 80% is likely produced by the $RHi_{min}$ parameter reflecting sub-grid scale RHi variance as mentioned above (Gettelman et al., 2010), which was set at the default value (80% RHi) for both simulations. The primary peak at 100% RHi is likely a result of the minimum threshold for heterogeneous ice nucleation being set at 120% as well as a sub-grid variability scaling factor 1.2 being considered (Wang et al., 2014a). Similar to 430-s averaged observations, IWC and Ni show steep increase (i.e., 3 – 4 orders of magnitude) from 40% to 100%. Increases of average IWC and Ni are seen in the simulations as RHi increases from 120% to 160%, differing from the decreasing trend seen in the observations. These higher values of IWC and Ni near 160% are possibly due to RHi reaching the homogeneous nucleation thresholds, where ice nucleation becomes more dependent upon temperature and updraft speed (Liu and Penner, 2005). Note that at this same point as IWC and Ni increase, there is a decrease in Di, which also suggests homogeneous nucleation in the model. For Di - RHi correlations, both simulations show similar results to the observations, with the maximum Di around 130% RHi and some large ice particles in the subsaturated conditions. The large variability of observed ice microphysical properties is also significantly underestimated in the model for ISS conditions. Standard deviations are 0.5 – 1 order of magnitude lower for IWC and Ni and a factor of 2 lower for Di compared with observations.

Comparing the correlations with $\sigma_w$ (Figure 14), the simulations show increasing IWC and Ni with higher $\sigma_w$, which agree with observations, although the increase of IWC and Ni are smaller in the simulations than the 430-s observations. The simulated Di is relatively constant with increasing $\sigma_w$, which differs from the slight positive correlation between Di and $\sigma_w$ in the observations. This slight positive Di - $\sigma_w$ correlation is likely due to the growth of ice particles as cirrus clouds evolve with continuous updrafts that supply excess water vapor above ice saturation, which was previously discussed in a cirrus cloud evolution analysis (Diao et al., 2013). The simulations may overlook this positive correlation due to several reasons, such as the lack of temporal resolution to resolve cirrus evolution in the growth phase, the lack of vertical velocity sub-grid variabilities (as discussed in Zhou et al. (2016)), and a dry bias (i.e., lower RHi) in the model (as discussed in Wu et al. (2017)).

Comparing the performance of two types of simulations, both CAM6-nudg and CAM6-free show bimodal distributions for IWC – RHi and Ni – RHi correlations, and they both show positive correlations for IWC – $\sigma_w$ and Ni – $\sigma_w$. This result indicates that the general trends in these correlations are statistically robust and less affected by sampling sizes and geographical locations. For correlations with RHi, the maximum IWC value in CAM6-nudg and CAM6-free is lower than the 430-s averaged observations by a factor of 25 and 100, respectively. The maximum Ni value in CAM6-nudg is similar to the 430-s averaged observations, while that value in CAM6-free is lower by a factor of 3. For correlations with $\sigma_w$, there are no significant differences for the maximum IWC between the two simulation types. The maximum Ni value in CAM6-nudg

and CAM6-free is higher than the 430-s averaged observations by a factor of 3 and 10, respectively. These results show that CAM6-nudg data, which are collocated with flight tracks, produce IWC and Ni values closer to the 430-s averaged observations than CAM6-free, possibly due to the variabilities of IWC and Ni in different geographical locations as shown in Figure 5.

## 4.3 Aerosol indirect effects

The effects of larger and smaller aerosols (i.e., $Na_{500}$ and $Na_{100}$) on ice microphysical properties are further examined for observations and CAM6-nudg data (Figures 15 and 16). Cloud fraction is calculated in each temperature – Na bin by normalizing the number of in-cloud samples with the total number of samples in that bin for both observations and simulations. For three cirrus microphysical properties (i.e., IWC, Ni and Di), positive correlations are seen in 1-Hz observations with respect to $Na_{500}$ and $Na_{100}$. In addition, higher $Na_{500}$ (>10 cm$^{-3}$) and $Na_{100}$ (>100 cm$^{-3}$) values are

associated with significant increases in cloud fraction. At -70ºC to -60ºC, higher IWC, Ni and cloud fraction are seen when $Na_{500}$ is observed, with positive correlations of IWC and Ni with respect to $Na_{500}$. This finding indicates that larger aerosols provide an effective pathway of ice particle formation for colder conditions. The higher IWC and Ni are only shown in much higher $Na_{100}$ (>100 cm$^{-3}$) between -70ºC and -60ºC, demonstrating that larger aerosols facilitate ice formation more effectively than smaller aerosols at this temperature range, possibly due to the activation of larger aerosols as INPs for

heterogeneous nucleation. Compared with 1-Hz observations, 430-s averaged observations show weaker correlations of IWC, Ni and Di with respect to Na. However, they do still show higher IWC and Ni between -70ºC and -40ºC associated with higher Na (i.e., $Na_{500} > 1$ cm$^{-3}$ and $Na_{100} > 30$ cm$^{-3}$).

The CAM6-nudg simulation shows increasing average IWC, average Ni and cloud fraction with increasing $Na_{500}$, consistent with the observations. But at temperatures below -60ºC, simulated IWC and Ni do not show a sudden increase with higher

$Na_{500}$ as shown in the observations. The simulated Di slightly decreases with increasing $Na_{500}$, differing from the increasing trend seen in observations. For aerosol indirect effect analysis based on $Na_{100}$, the comparison results are similar to $Na_{500}$, that is, CAM6-nudg simulation is able to represent positive correlations of IWC, Ni and cloud fraction with respect to $Na_{100}$. However, CAM6-nudg simulation shows smaller (larger) increase of IWC (Ni) at very high Na (i.e., $Na_{500} > 1.6$ cm$^{-3}$ and $Na_{100} > 30$ cm$^{-3}$) compared with the 430-s averaged observations. The model also misses positive correlations between Di

and Na seen in both 1-Hz and 430-s averaged observations.

## 5 Discussion and conclusions

In this study, we investigate the statistical distributions of cirrus cloud microphysical properties (i.e., IWC, Ni, and Di) as well as several key controlling factors (i.e., temperature, RHi, $\sigma_w$ and Na) using a comprehensive in situ observational dataset and GCM simulations. Regional variations of cirrus cloud microphysical properties are examined for six latitudinal

regions in two hemispheres. Two types of CAM6 simulations are evaluated, i.e., nudged and free-running simulations.

Regarding the regional variations in 1-Hz observations, the highest and lowest IWC values were observed in NH midlatitude and SH midlatitude, respectively, while the polar regions show the lowest Ni and highest Di at warmer conditions (i.e., -55ºC to -40ºC) (Figure 5). The hemispheric differences between NH and SH midlatitudes indicate a possible role of anthropogenic aerosols and/or land-sea contrast in controlling ice microphysical properties. Thermodynamic and dynamic conditions can also affect nucleation mechanisms. For example, the tropical regions show the highest IWC and Ni at temperatures below -55ºC possibly due to convection anvils with the droplet freezing from down below or homogeneous nucleation in gravity waves generated by convection. This feature is corroborated by the fact that tropical regions show the highest RHi values for both clear-sky and in-cloud conditions (Figure 6), while the midlatitude and polar regions show fewer samples exceeding the homogeneous nucleation threshold. The higher RHi values in tropics are likely contributed by higher updrafts (indicated by higher $\sigma_w$ in Figure 9). These results demonstrate the important roles of these controlling factors on cirrus clouds at different latitudinal and temperature ranges.

Evaluating the model simulations of cirrus microphysical properties, different model performance results are seen in different regions. For example, simulations underestimated the IWC in NH (Figure 5), possibly due to model dry bias to form ice clouds (as discussed in Wu et al. (2017)) and/or smaller aerosol indirect effects on IWC in the simulations (Figures 15 and 16). Differences in the particle size distribution, such as lower number density of larger particles (> 1000 μm) in the simulation (Figure 4 a), may also contribute to the underestimation of IWC by the simulation. All the comparison results on IWC, Ni and Di are only applicable to the size range being evaluated ($\geq 62.5$ μm). For RHi distributions, both simulations represent a similar peak position at ice saturation for in-cloud RHi PDFs compared with observations but CAM6-nudg underestimates the frequency and magnitude of ISS for clear-sky condition. For $\sigma_w$ distributions, simulations represent similar regional variations of $\sigma_w$ compared with observations, with $\sigma_w$ decreasing from lower to higher latitudes. The model performs well for representing the effects of RHi and $\sigma_w$ on ice microphysical properties, specifically for showing the maximum IWC and Ni at 100% RHi, and the positive correlations with $\sigma_w$. Some differences include the simulated average IWC and Ni showing a secondary peak position at 80% RHi, likely due to the minimum RHi threshold used in the model parameterization. Both simulation types show similar correlation trends of ice microphysical properties with respect to RHi and $\sigma_w$. CAM6-nudg performs better for representing IWC and Ni magnitudes than CAM6-free, possibly due to better collocation between CAM6-nudg and observations.

For aerosol indirect effects, the simulations underestimate IWC, Ni, Di as well as cloud fraction at colder conditions (< -60ºC) when larger aerosols exist, indicating that the effectiveness of larger aerosols is underestimated at the colder conditions. The observations also show higher Di than simulations by a factor of 3 – 4 at warmer temperatures (-50ºC to -40ºC), indicating misrepresentation of ice particle growth and/or sedimentation in the simulations. In addition, the IWC, Ni and Di in 430-s averaged observations show increase at higher $Na_{500}$ (>1 cm$^{-3}$) and $Na_{100}$ (>30 cm$^{-3}$), while simulations only show significant increase of Ni. This result indicates that aerosol indirect effects may be underestimated especially for higher Na values. It is possible that small ice crystals < 62.5 μm may have formed under high Na but are excluded due to the size constraint. Additionally, because INP activation is highly dependent upon temperature, we acknowledge the limitation

of using Na$_{500}$ to indicate INP concentrations. The assumption of ice mass and dimension relationship from Brown and Francis (1995) may also lead to uncertainties due to various ice habits. These caveats call for more investigation on small ice measurements, INP measurements at temperature ≤ -40°C, and measurements of various ice habits.

Overall, the global-scale observational dataset used in this study provides statistically robust distributions of cirrus cloud microphysical properties, which can be used to evaluate the effects of thermodynamics, dynamics and aerosols on cirrus clouds in a global climate model. Extending from previous studies that investigated climate model sensitivity to individual cirrus cloud controlling factors, i.e., w (Shi and Liu, 2016), RHi (D'Alessandro et al., 2019), water vapor (Wu et al., 2017), and aerosols (Wang et al., 2014a), this study provides an analysis of all factors. In addition, further attention was given towards evaluating these factors in the simulations based on geographical locations. For both observations and simulations, higher ice supersaturations and stronger vertical motions are shown in tropical and midlatitude regions, which possibly lead to increased homogeneous nucleation and convection-generated cirrus, consistent with higher IWC and Ni and lower Di in these regions compared with polar regions. In addition, underestimating aerosol indirect effects in the simulations likely contributes to the underestimation of IWC in the NH. Even though small ice particles (< 62.5 μm) are excluded in this study, correlations between ice microphysical properties and these key controlling factors are still clearly seen in the observation dataset. In addition, using two methods that compare observations on the horizontal scales of 230 m and 100 km with simulations, both methods show similar signs for model biases of IWC, Ni and Di, while smaller model biases are seen when comparing against the coarser resolution observations. This study underscores the importance of correctly representing the thermodynamic, dynamic and aerosol conditions in climate models at various regions, as well as accurately simulating their correlations with ice microphysical properties. Failing to do so may result in biases of cirrus cloud microphysical properties depending on different regions and temperatures, leading to biases in cirrus cloud radiative effects on a global scale.

**Data Availability**

Observations from the seven NSF flight campaigns are accessible at https://data.eol.ucar.edu/.

**Author contributions**

R. Patnaude and M. Diao contributed to the development of the ideas, conducted quality control to aircraft-based observations, and wrote the majority of the manuscript. R. Patnaude contributed to all model simulations and the subsequent data analysis. X. Liu and S. Chu provided expertise on the set-up of CAM6 model simulations and provided input to the analysis of simulation data.

## Competing interests

The authors declare that they have no conflict of interest.

## Acknowledgments

450 R. Patnaude and M. Diao acknowledge funding support from U. S. National Science Foundation grants AGS-1642291 and OPP-1744965. R. Patnaude also acknowledges support from the San Jose State University Walker Fellowship. For funding support in 2016 and 2018 summer, M. Diao acknowledges the NCAR Advanced Study Program (ASP) Faculty Fellowship. X. Liu and S. Chu acknowledge the support of the National Science Foundation under grant AGS-1642289 / 2001903. We would like to acknowledge the NCAR/Earth Observation Laboratory flight teams from the seven flight campaigns: 455 START08, HIPPO, PREDICT, DC3, CONTRAST, TORERO, and ORCAS. For in situ observations of water vapor by the VCSEL hygrometer, field support, calibration and QA/QC was conducted by M. Diao, J. DiGangi, M. Zondlo, and S. Beaton. Additional appreciation is given to Jorgen Jensen, Chris Webster, and Christina McCluskey for helpful discussions.

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

**Table 1.** Descriptions of seven NSF flight campaigns conducted with the NSF/NCAR G-V research aircraft.

| Acronym | Field Campaign | Time | Lat, Lon | Region* | Flight hours | In-cloud hours | Clear-sky hours |
|---|---|---|---|---|---|---|---|
| START08[a,b] | Stratosphere-Troposphere Analyses of Regional Transport | April – June 2008 | 26°N – 62°N, 117°W – 86°W | NM, NP | 84 | 2 | 52 |
| HIPPO[c-f] | HIAPER Pole-to-pole Observations deployments 2 – 5 | Oct – Nov 2009; Mar – Apr 2010; Jun – July 2011; Aug – Sept 2011 | 87°N – 67°S, 128°E – 90°W | A | 333 | 7 | 111 |
| PREDICT[g] | PRE-Depression Investigation of Cloud Systems in the Tropics | Aug – Sept 2010 | 10°N – 28.5°N, 86°W – 37°W | NT | 105 | 25 | 66 |
| DC3[h] | Deep Convective Clouds and Chemistry Project | May – June 2012 | 25°N – 42°N, 106°W – 80°W | NM | 144 | 23 | 54 |
| TORERO[i] | Tropical Ocean tRoposphere Exchange of Reactive halogen species and Oxygenated voc | Jan – Feb 2012 | 42°S – 14°N, 105°W – 70°W | NT, ST, SM | 125 | 2 | 52 |
| CONTRAST[j] | CONvective TRansport of Active Species in the Tropics | Jan – Feb 2014 | 20°S – 40°N, 132°E – 105°W | NM, NT, ST | 116 | 23 | 48 |
| ORCAS[k] | The $O_2/N_2$ Ratio and $CO_2$ Airborne Southern Ocean (ORCAS) Study | Jan – Mar 2016 | 75°S – 18°S, 91°W – 51°W | SM, SP | 95 | 1 | 40 |

*: N, northern hemisphere; S, southern hemisphere; T, tropics; M, midlatitude; P, polar regions; A, all regions.

[a,b] (UCAR/NCAR, 2009, 2019 a); [c-f] (UCAR/NCAR, 2019 b-e); [g] (UCAR/NCAR, 2019 f); [h] (UCAR/NCAR, 2018 a); [i] (UCAR/NCAR, 2019 g); [j] (UCAR/NCAR, 2018 b); [k] (UCAR/NCAR, 2018 c). Full citations of each data set are included in the reference list.

**Table 2.** Ranges of meteorological conditions and ice microphysical properties for in situ 1-Hz observations, 430-s averaged observations, CAM6-nudg and CAM6-free data used in this study.

| | In situ observations | 430-s averaged observations | CAM6-nudg | CAM6-free |
|---|---|---|---|---|
| T (ºC) | -78 – -40 | -77 – -40 | -75 – -40 | -89.9 – -40 |
| P (Pa) | 12,389 – 53,137 | 37,778 – 53,410 | 12,300 – 53,446 | 12,300 – 53,100 |
| RHi (%) in-cloud (clear sky) | 0.99 – 175.1 (0.3 – 174.9) | 0.3 – 153.7 (0.3 – 134.6) | 0.8 – 159.8 (0.05 – 107.6) | 0.003 – 257.2 (0.001 – 181.4) |
| IWC (g/m$^3$) | 0.00004 – 23.31 | 1e-7 – 11.58 | 1e-7 – 0.12 | 1e-7 – 0.16 |
| Ni (#/L) | 0.039 – 542.15 | 9.6e-5 – 188.7 | 1e-4 – 207.04 | 1e-4 – 516.7 |
| Di (μm) | 62.5 – 3200 | 62.5 – 2175 | 62.5 – 2062 | 66.7 – 2556 |

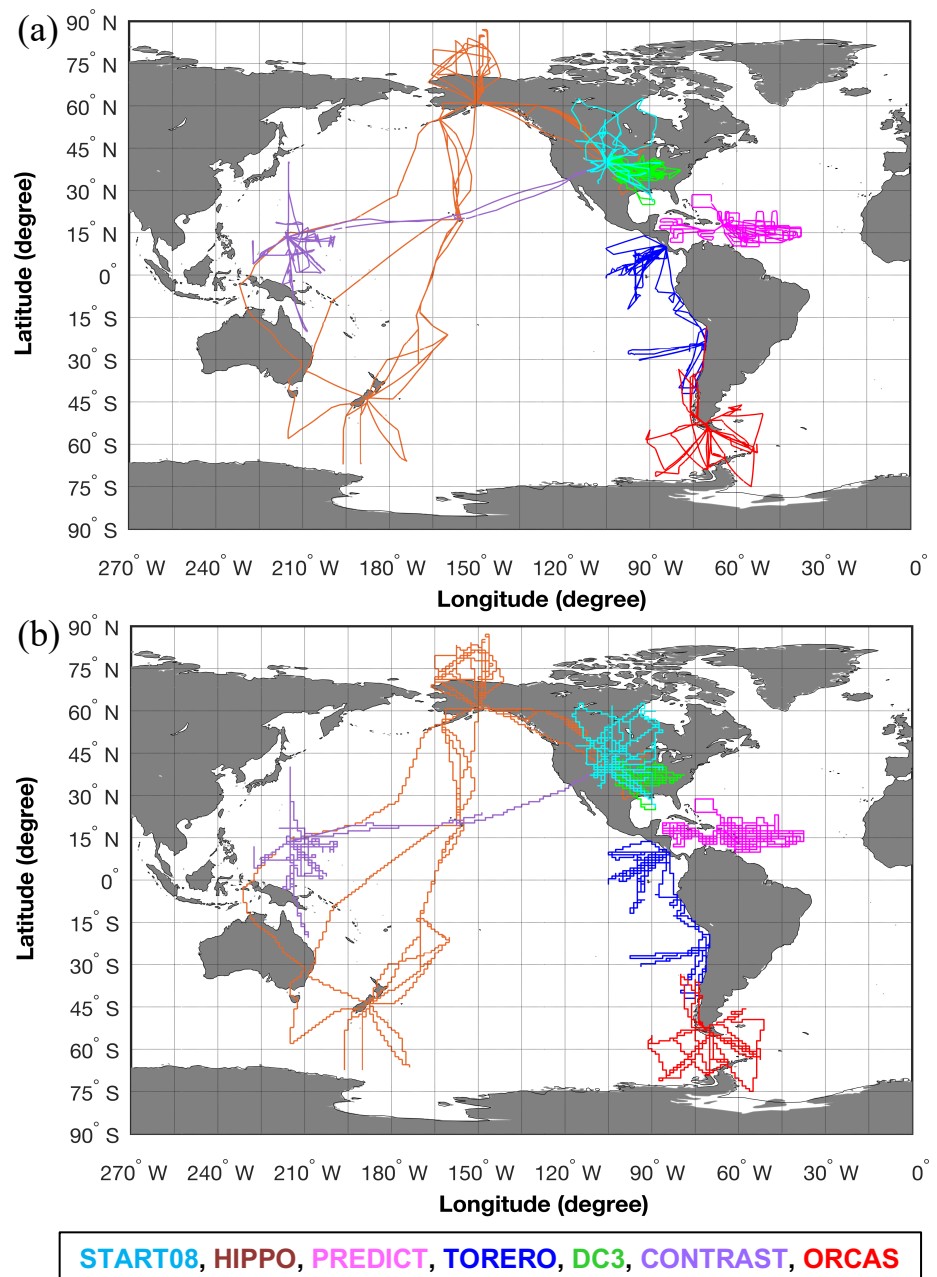

**START08**, **HIPPO**, **PREDICT**, **TORERO**, **DC3**, **CONTRAST**, **ORCAS**

**Figure 1.** Global maps of flight tracks representing the seven campaigns in this study for (a) in situ observations and (b) CAM6-nudg. Colors denote different campaigns.

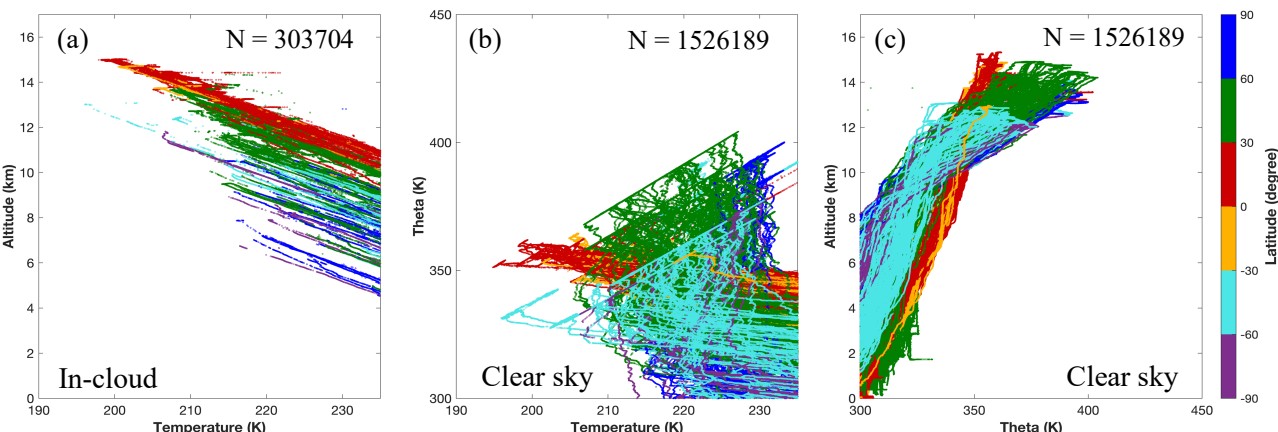

**Figure 2.** (a) Vertical profiles of temperature, (b) potential temperature vs. temperature, and (c) vertical profiles of potential temperature based on in situ observations at temperatures ≤ -40ºC. Number of samples (N) for 1-Hz observations is shown in the figure legend. Colors denote six latitudinal regions.

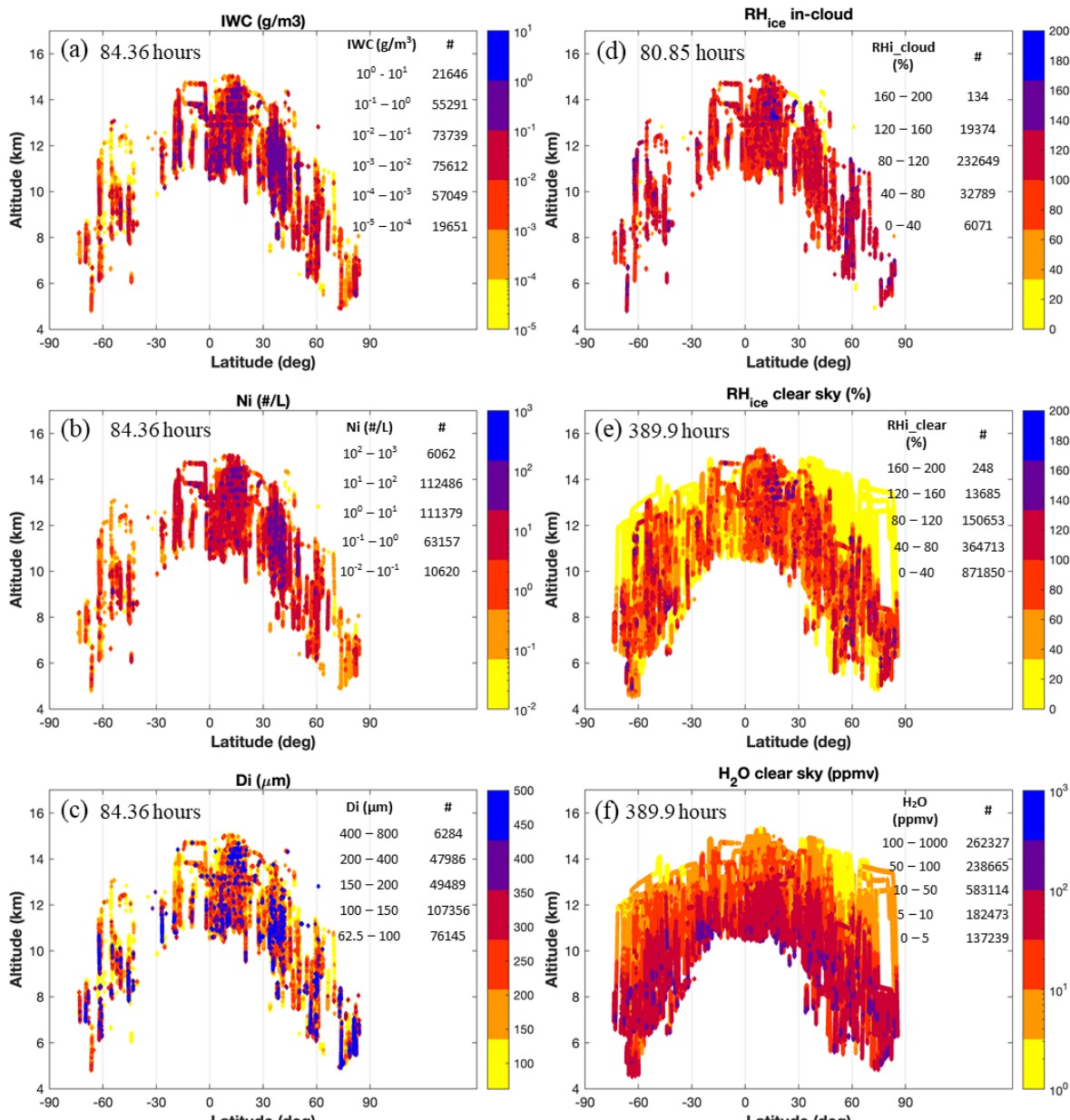

**Figure 3.** Latitude and altitude distributions of (a) IWC, (b) Ni, (c) Di, (d) in-cloud RHi, (e) clear-sky RHi, and (f) clear-sky water vapor volume mixing ratio at temperatures $\leq$ -40ºC. Total measurement hours and number of samples for given intervals are shown for each variable. Note that the measurement ranges shown in the upper right corner are not the full

ranges (see Table 2 for the full ranges).

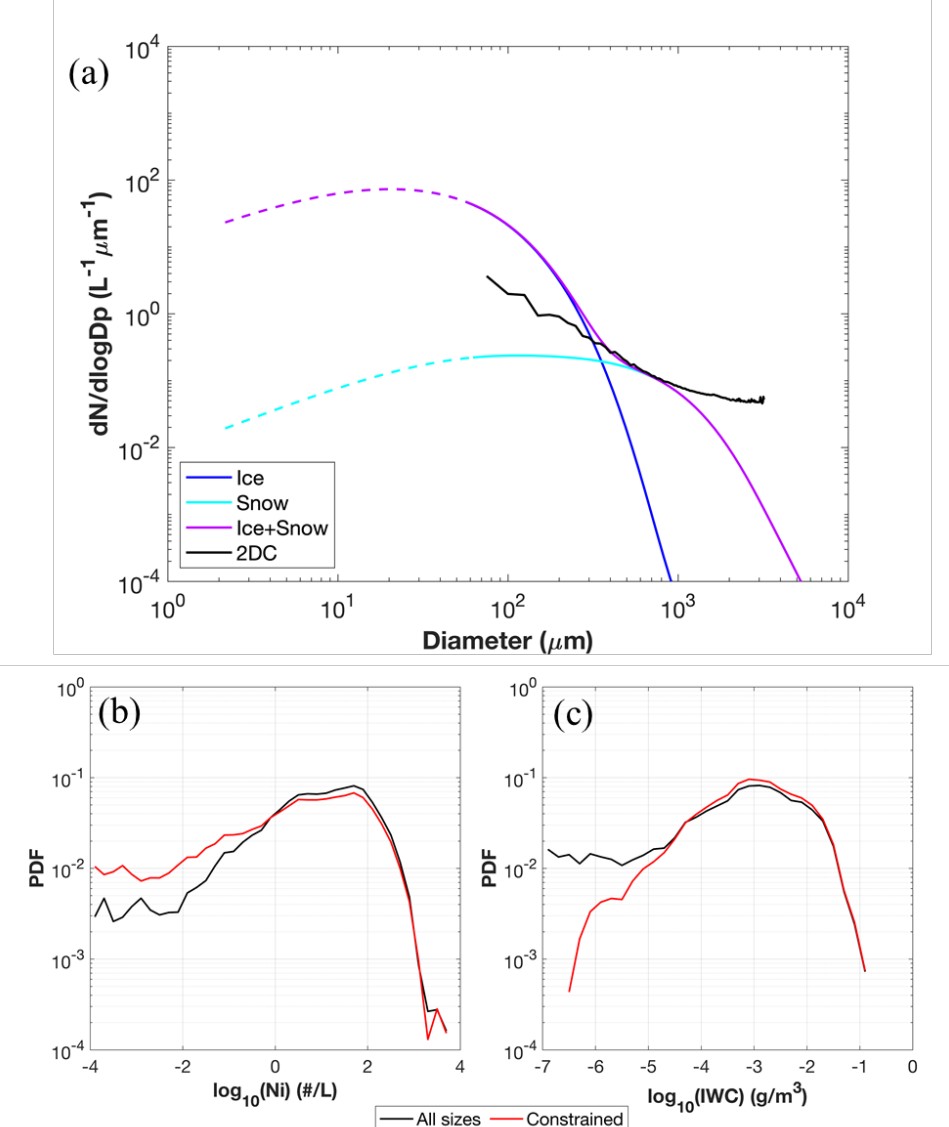

**Figure 4.** (a) Observed size distribution (black line) and reconstructed size distributions from simulated ice (blue) and snow

(cyan). Size truncations to diameters < 62.5 µm (dashed lines) are shown for simulated hydrometeors, while the remaining particles (≥ 62.5 µm) (solid lines) are used for comparisons with observations. Size distributions for combined ice and snow in the simulations (purple) are also shown before and after the size restriction. (b) and (c): PDFs of Ni and IWC in the simulation before and after size truncation.

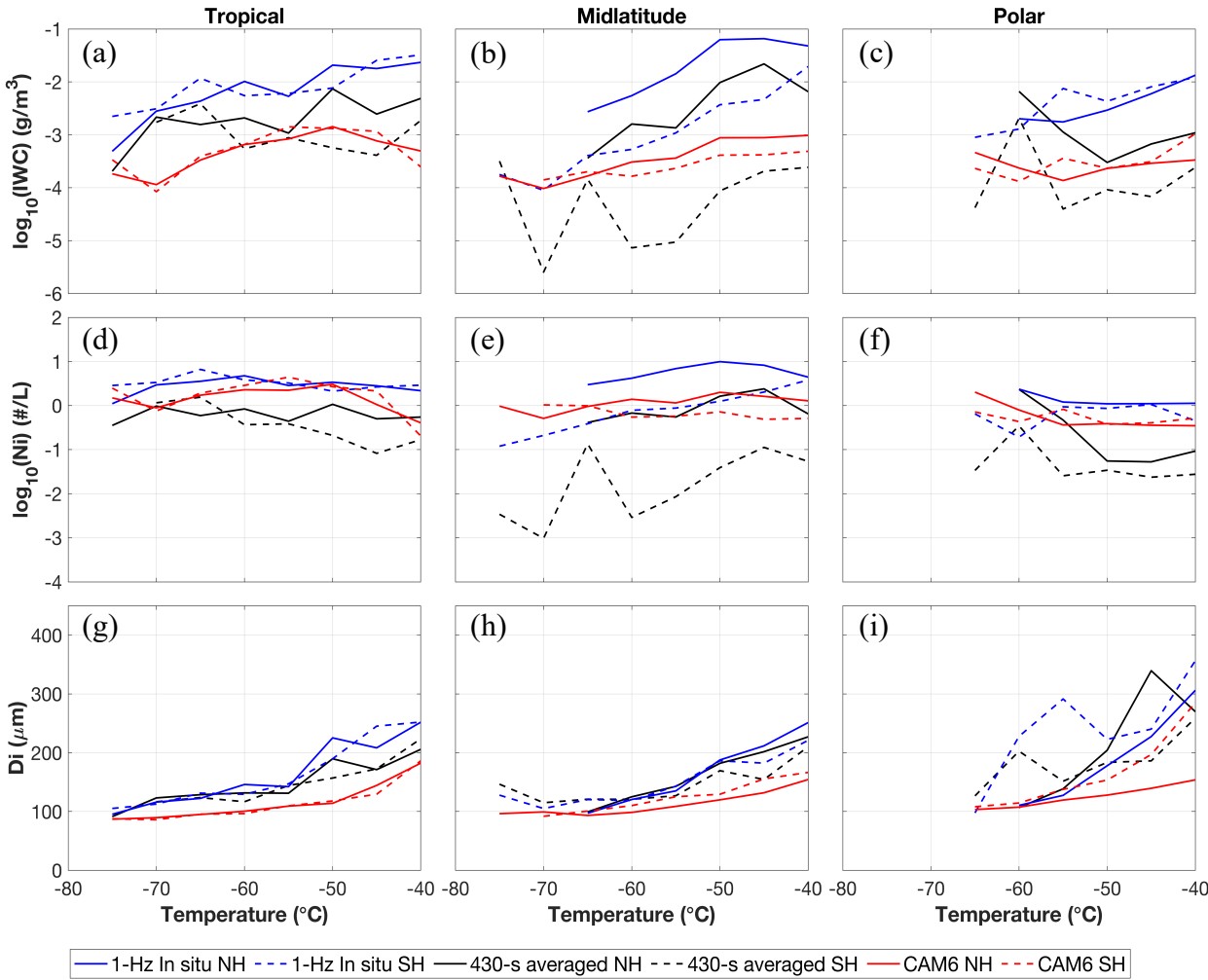


**Figure 5.** Geometric means of (a – c) IWC and (d – f) Ni, as well as (g – i) linear averages of Di at 5ºC temperature intervals between -80ºC and -40ºC, compared between 1-Hz in situ observations (blue lines), 430-s averaged observations (black lines) and CAM6-nudg (red lines). Observed and simulated microphysical properties are binned by six latitudinal regions, where NH is denoted by solid lines, and SH is denoted by dashed lines. The number of samples for 1-Hz observations at temperatures ≤ -40ºC in the northern (southern) hemisphere tropical, midlatitude and polar regions are 173930 (15569), 100615 (3809), and 6704 (2606), respectively. The number of samples for 430-s averaged observations in these regions are 355082 (40683), 233546 (26850) and 24083 (10252), respectively. The number of samples for CAM6-nudg data in these regions are 3241592 (653110), 2052353 (590503) and 478844 (209662), respectively.

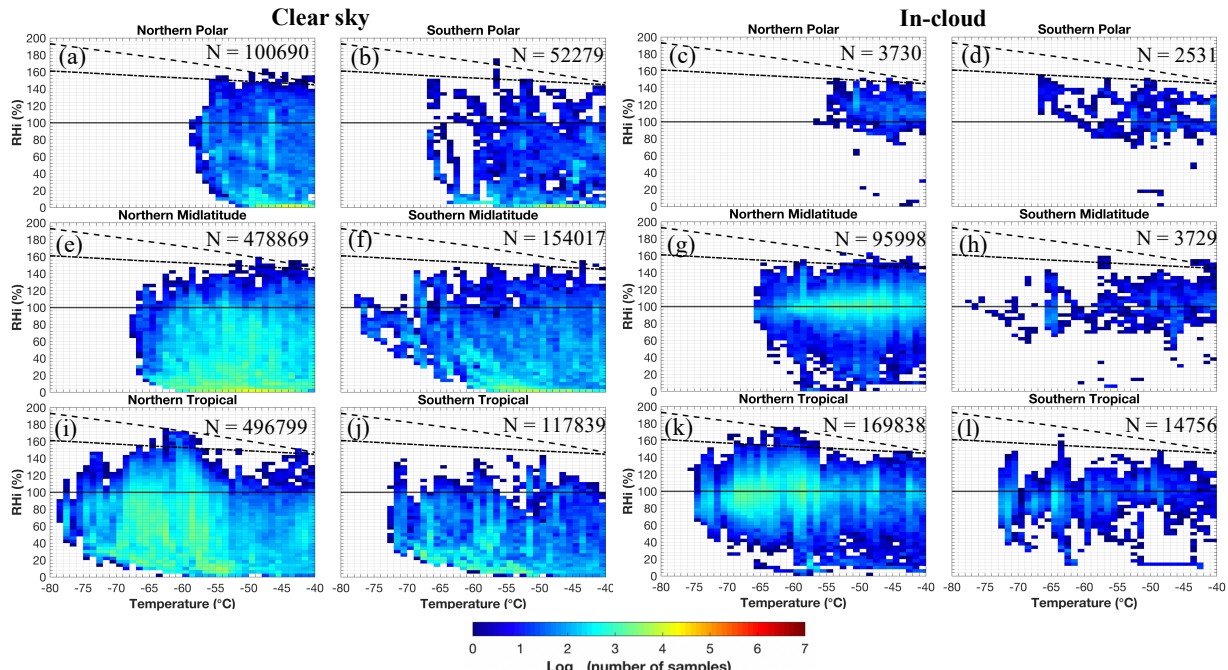

**Figure 6.** Distributions of RHi at various temperatures and geographical locations from in situ observations under (left two columns) clear-sky and (right two columns) in-cloud conditions. Solid and dashed black lines represent ice and liquid saturation, calculated based on saturation vapor pressure with respect to ice and liquid from Murphy and Koop (2005), respectively. Dash-dotted line denotes the homogeneous freezing threshold for 0.5 μm aerosols based on Koop et al. (2000).

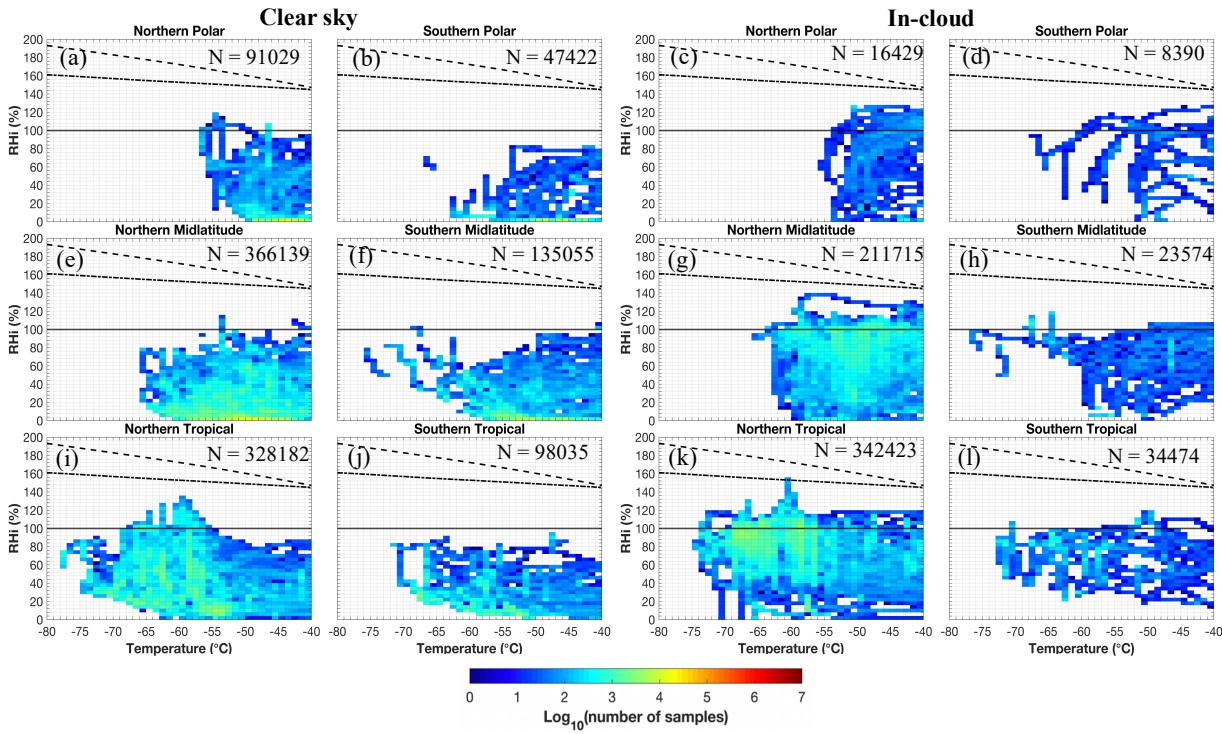

**Figure 7.** Similar to Figure 6 but for 430-s scale.

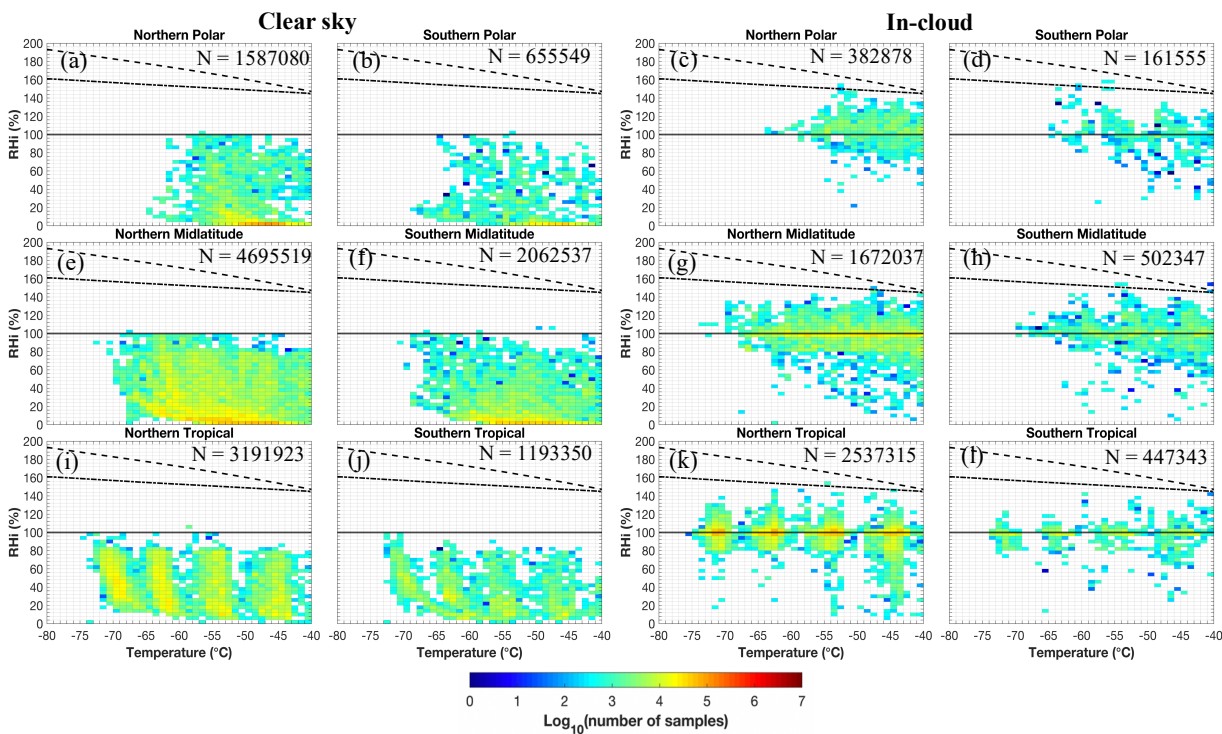

**Figure 8**. Similar to Figure 6 but for CAM6-nudg data. RHi values for simulations are calculated using simulated specific humidity and temperature, based on the equation of saturation vapor pressure with respect to ice from Murphy and Koop (2005).

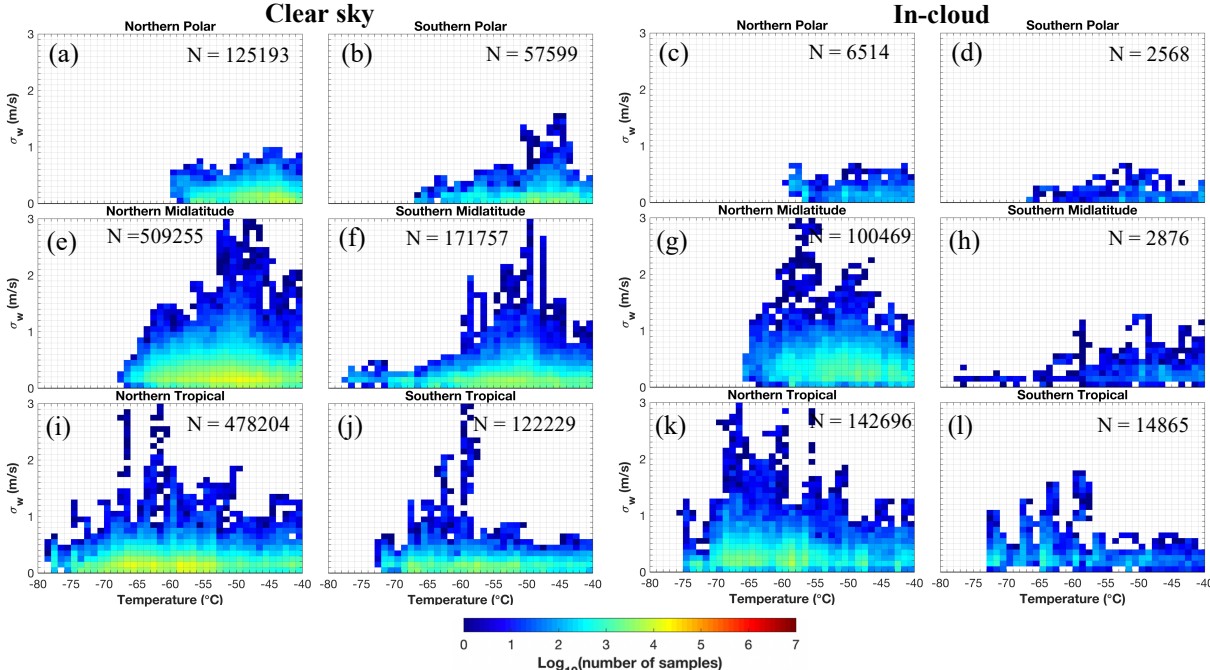

**Figure 9.** Distributions of $\sigma_w$ calculated for every 40 seconds using the 1-Hz observations under (left two columns) clear-sky and (right two columns) in-cloud conditions.

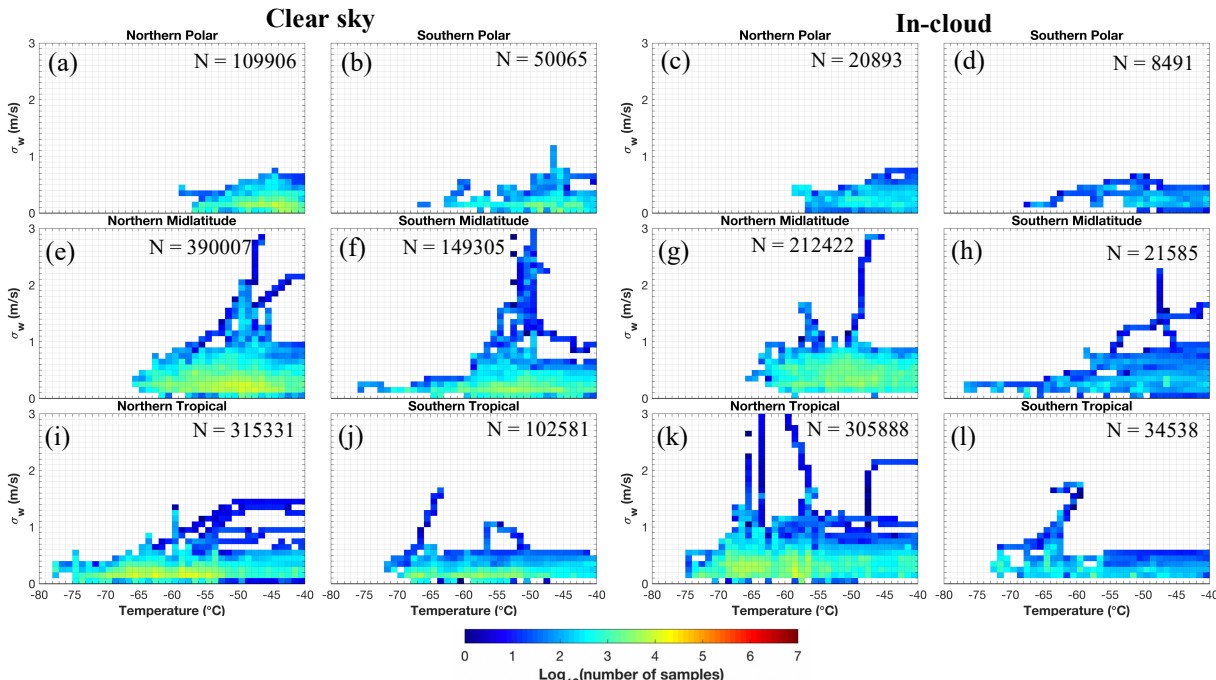

**Figure 10.** Similar to Figure 9, but for 430-s averaged observations.

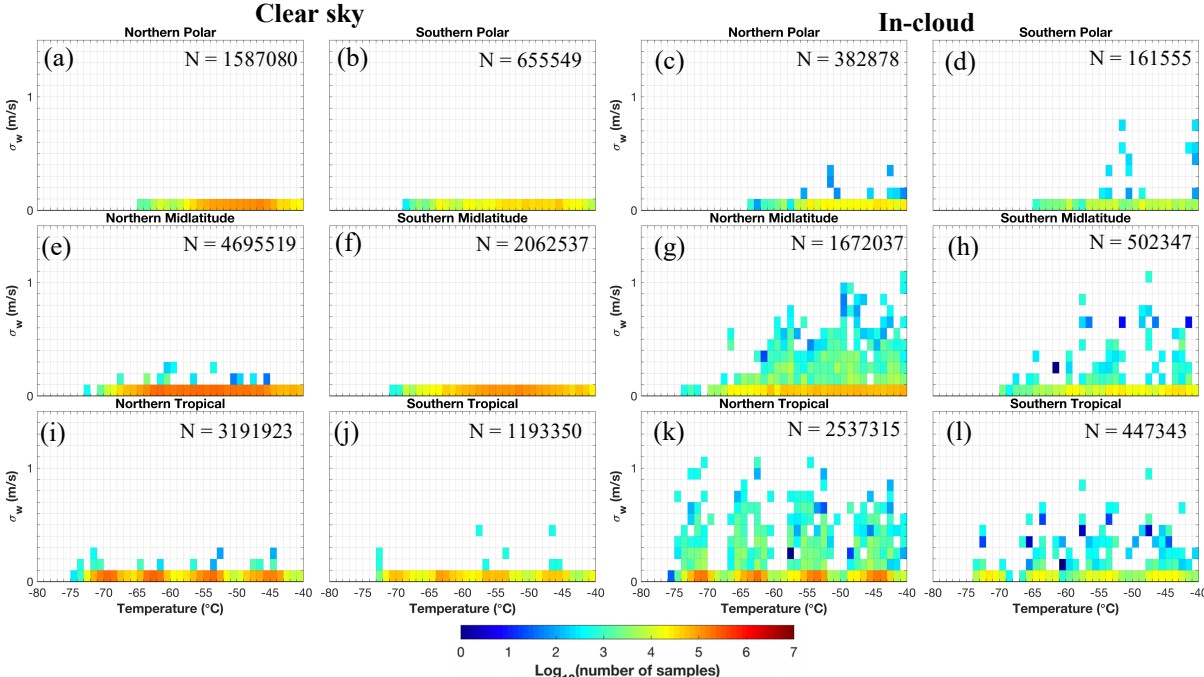

790

**Figure 11**. Similar to Figure 9 but for the CAM6-nudg data.

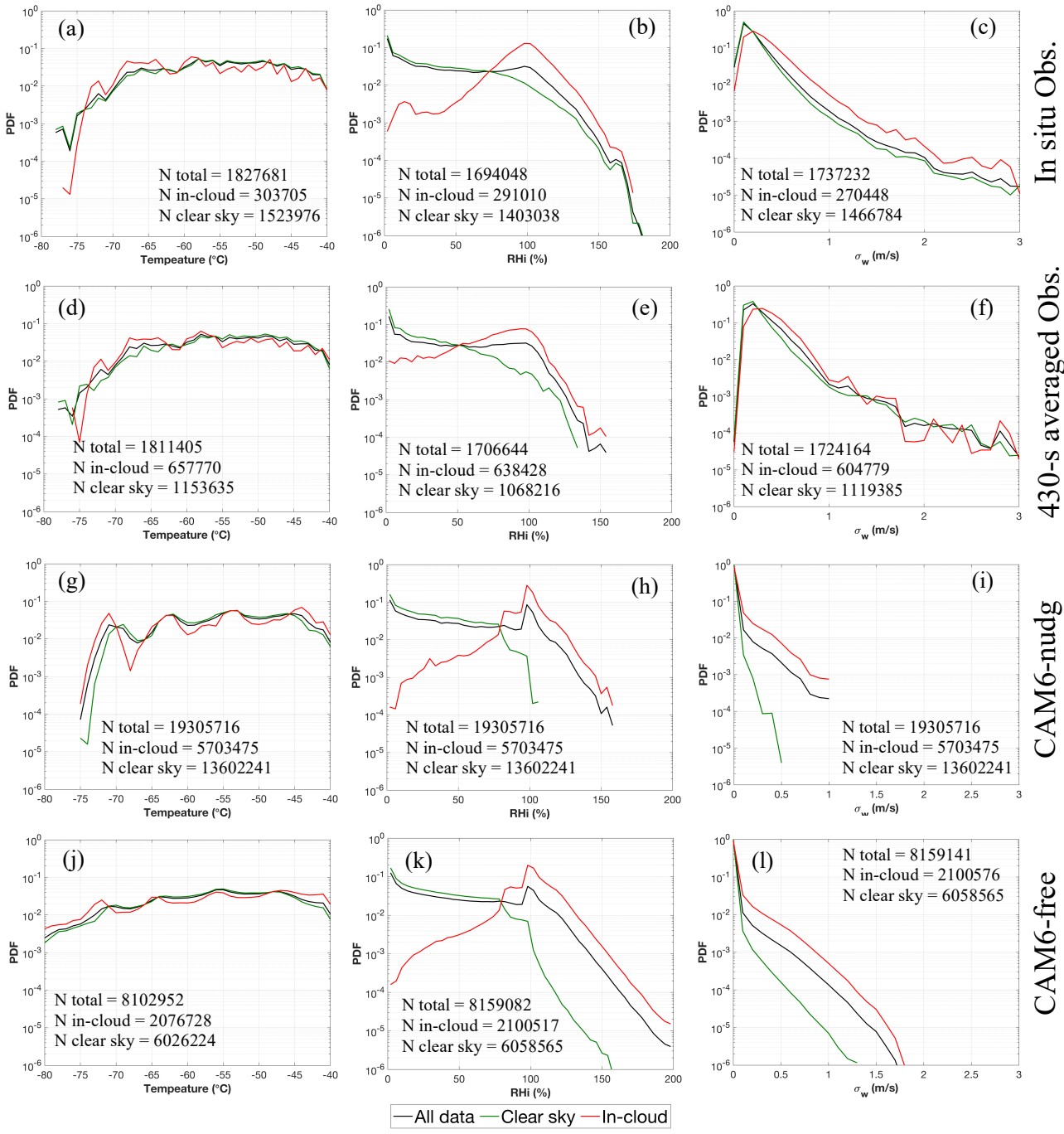

**Figure 12.** Probability density functions (PDFs) for (left column) temperature, (middle column) RHi and (right column) $\sigma_w$, compared among (a– c) 1-Hz observations, (d – f) 430-s averaged observations, (g – i) CAM6-nudg and (j - l) CAM6-free data. Note that $\sigma_w$ in (c) is calculated for every 40 seconds.

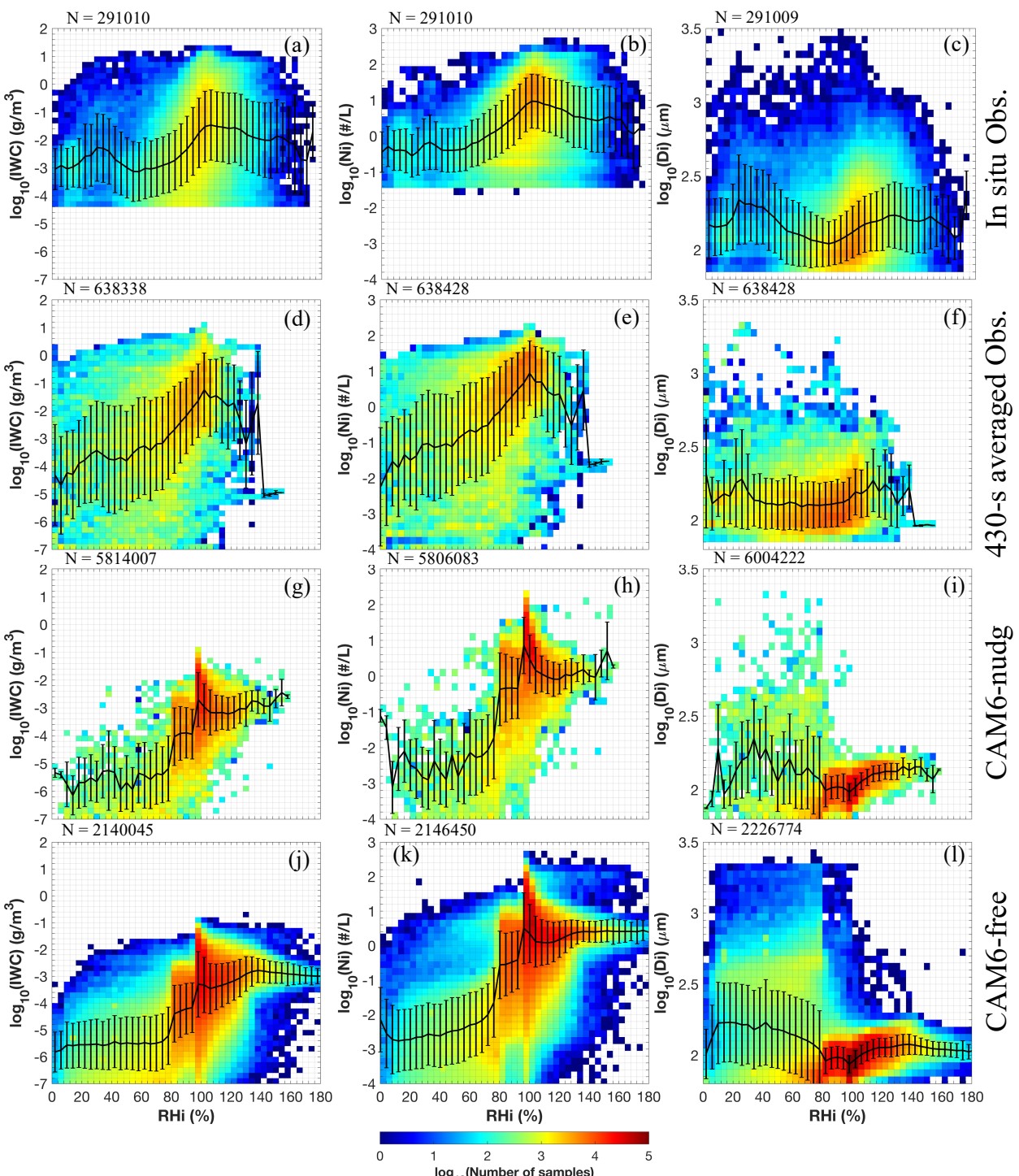

**Figure 13.** Correlations between RHi and in-cloud IWC, Ni and Di (columns 1 – 3, respectively), compared among (a – c) in situ 1-Hz observations, (d – f) 430-s averaged observations, (g – i) CAM6-nudg, and (j – l) CAM6-free data. Black lines and whiskers denote geometric means and standard deviations, respectively.

800

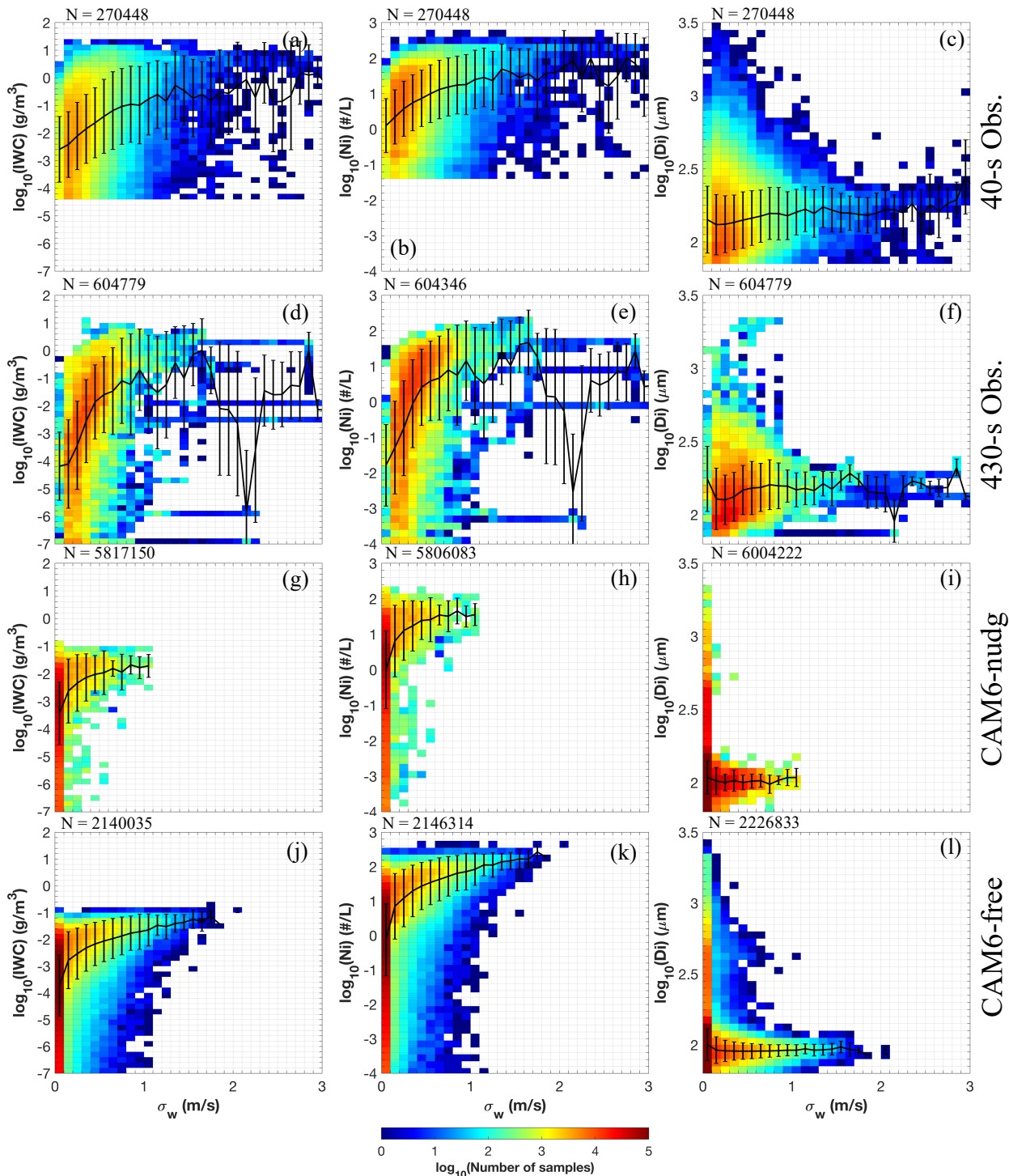

**Figure 14.** Similar to Figure 13 but for correlations with σ$_w$. Note that σ$_w$ of observations are calculated for (a – c) every 40 seconds and (d – f) every 430 seconds using the 1-Hz observations.

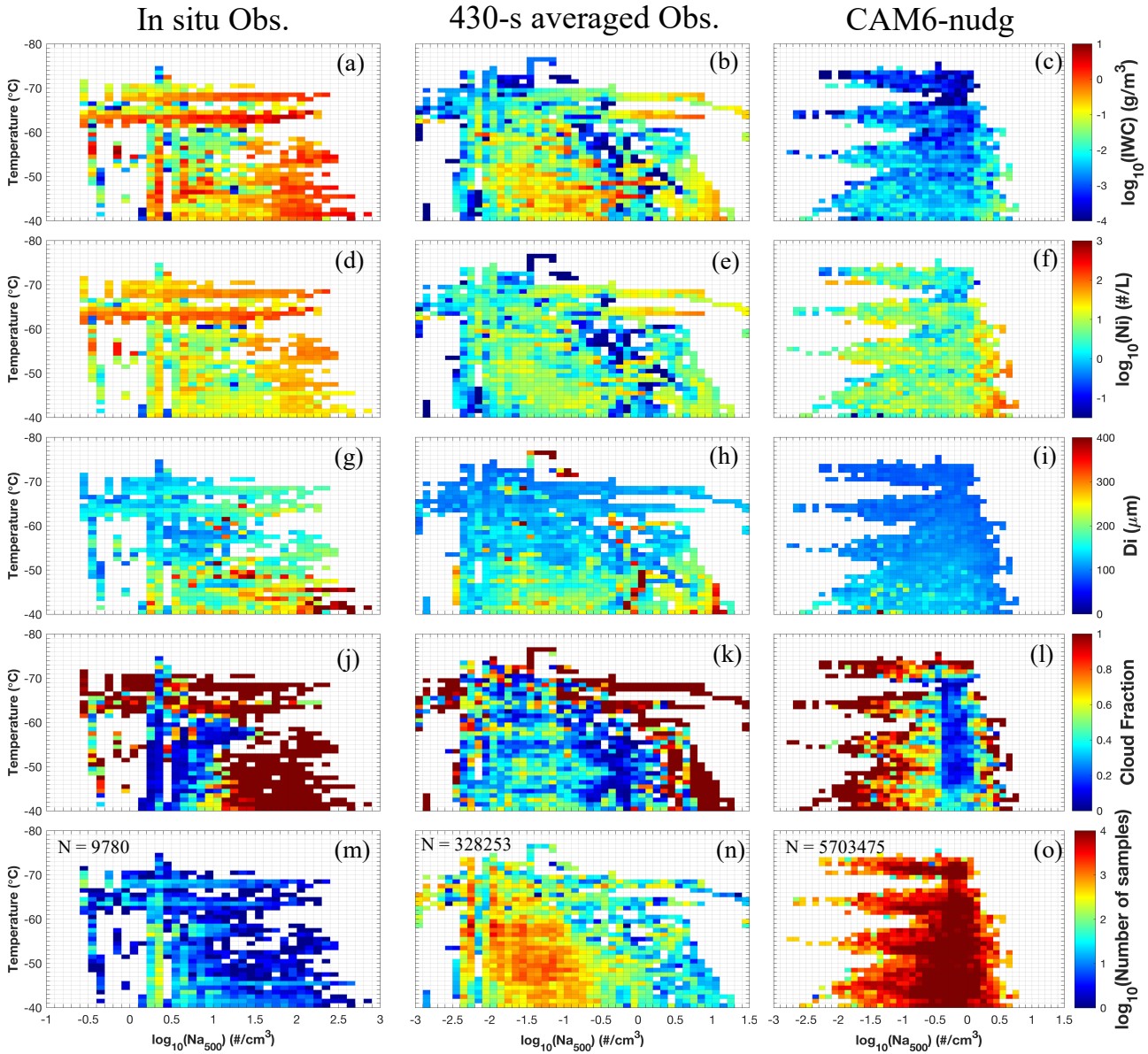

**Figure 15.** Aerosol indirect effects from logarithmic-scale Na$_{500}$ on (a – c) IWC, (d – f) Ni, (g – i) Di, and (j – l) cloud fraction, compared among 1-Hz observations (left column), 430-s averaged observations (middle column) and CAM6-nudg data (right column). Number of samples of each bin is shown in the bottom row (m – o). Cloud fraction is calculated as the number of in-cloud samples over the total number of samples for a given temperature and Na bin.

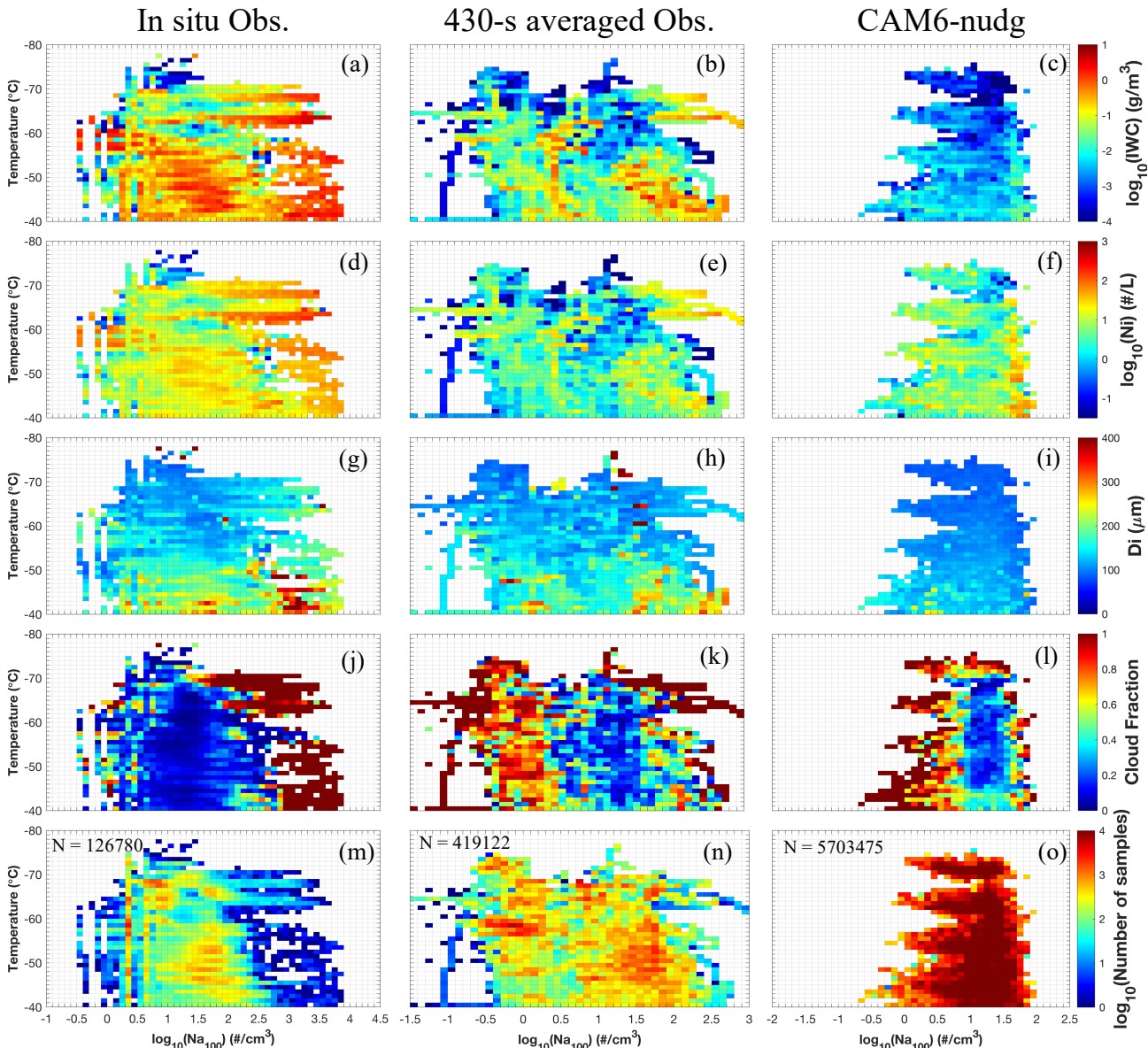

810    **Figure 16.** Similar to Figure 15, but examined for $\log_{10}(Na_{100})$.