# Peer review of "Effects of Thermodynamics, Dynamics and Aerosols on Cirrus Clouds Based on In Situ Observations and NCAR CAM6 Model"

_Atmospheric Chemistry and Physics, 2020_

## Referee Comment (RC1) · Andrew Gettelman (Referee) · 24 Aug 2020

Review of "Effects of Thermodynamics, Dynamics and Aerosols on Cirrus Clouds Based on In Situ Observations and NCAR CAM6 Model" by Patnaude et al.

Review by Andrew Gettelman, NCAR

In general this is a very well written cutting edge analysis of a comparison between observations of upper tropospheric ice from aircraft and an advanced large scale global climate model. However, I have some questions on how the analysis was conducted and the sampling of the models and observations. The work is very cutting edge in

the detail of the comparisons, but probably needs a few more pieces of description, and more information perhaps to back up the analysis. I think the manuscript needs some substantial revisions before it will be acceptable for publication in Atmospheric Chemistry and Physics.

1. In general, I am concerned that the comparison between the model and observations is not sampling them the same way. It does not seem as if the observations are averaged over a model grid box length before actually reporting values. This should be clarified.

2. But fundamentally, when this is done, you get a distribution of individual observations which make up a size distribution. In CAM6, it reports a single mass and number for clouds in a grid box. But this represents a size distribution itself. These size distributions can be compared: it may be that the mass and number is distributed differently than the observations. Gettelman et al 2020, in Press, JGR illustrates this method. You can plot the size distributions from the model by reconstructing the distributions from the model equations.

Gettelman, A, Charles G. Bardeen, C. S. McCluskey, and E. Jarvinen. "Simulating Observations of Southern Ocean Clouds and Implications for Climate." J. Geophys. Res., 2020. https://doi.org/10.1029/2020JD032619.

3. This leads to my next point. The lack of observations below 62 microns diameter means there may be a significant amount of missing ice in the observations. Have you used the distribution functions to truncate the model size, number and mass to reflect this? If not, it's going to make the comparison even worse. If so, there may be an issue just with the size distributions themselves (see earlier point).

4. I find it hard to imagine that a climate model can produce 30x less ice than observed and still produce reasonable radiative fluxes, particularly for the Outgoing Longwave Radiation. Maybe I am wrong. It's even worse if the model has small ice that is not seen in the observations, but it's explainable if you have removed all the small crystals

already. But since you are actually deriving IWC from a distribution of particles, it seems possible that the assumptions you are making may be very wrong (i.e. the number/size v. IWC relationship). What is the uncertainty here? See specific comment below.

5. Also, there is very little commentary on whether the nudged or free running reproduces observations better. At least not in the conclusions. I think there were some vague comments about the simulations being 'similar'. Is there no appreciable difference?

Specific Comments:

Page 2, L43: there are IWC observations and compilations that go back decades. Much of it by Heymsfield. Please cite some earlier work.

Page 2, L51: Fu and collaborators and Mitchell and collaborators have done some work with CALIPSO that might be relevant here, particularly for occurrence and particle size. IWC from CALIPSO is harder.

Page 4, L107: CAM6-nudg? Typo? Seems like you are using this throughout, but I don't see it defined.

Page 4, L110: Are you going to test the impact of this assumption? Does CAM6 have a similar or close relationship between IWC and number/size?

Page 4, L113: since CAM6 has 2 moments and a distribution, wouldn't it be wise to show the size distribution from observations (not just the mean diameter) and from CAM to see if there are biases in the shape or in parts of the size distribution?

Page 4, L119: 1hz is only maybe 200m horizontal. How can that be compared with a global model at 100km resolution?

Page 4, L121: does CAM6 also use Murphy and Koop? Could that be an issue (probably not).

Page 4, L123: whereas —> where

Page 4, L128: you might note that some latitudes have some very different regimes.

Page 5, L138: I'm not sure I found where this size cutoff is noted. Is it wise to proceed with only half the size distribution? Seems like that would strongly affect how forceful you can make the conclusions. It also means the model needs to be sampled carefully.

Page 5, L140: so I assume this is then a sampling bias to your observational data set?

Page 5, L150: it would be also worth noting the most ice relevant adjustments in CAM6: the use of hoose et al mixed phase ice nucleation, and shi ET al modifications for preexisiting ice.

Page 6, L169: this is a pretty substantial limitation and should be noted much earlier in the text and even have caveats in the abstract. Showing model size distributions I think is critical here. Are you calculating simulated IWC without the small particles? Would that skew the results?

Page 6, L172: as noted above I think the method here is critical. Please describe it. I would feel more comfortable if you show he size distribution by deriving it from the mu and lambda of the gamma distributions, to understand the truncation issue.

Page 6, L177: I suggest that this is a major limitation of correlating Na500 with INP at the temperatures you are working with: INP activation is a strong function of temperature.

Page 6, L187: do you want to comment why here? SH is oceanic, NH is contintential.

Page 6, L190: tropical regions with colder temps might have more small crystals. Would this affect the CAM v. OBS results? I'm concerned you have not filtered the cam results by truncating size distributions.

Page 6, L194: I find the fact that you can get the OLR right with 10x less ice a little bit strange. Does ice mass not mater as all? Or does the di change compensate? Or

is the underestimation a product of the Observations really missing a lot of small ice? These don't seem to work the same way. So I am concerned that you are comparing 1hz IWC v. 100km IWC and this is producing anomalous results.

Page 7, L208: the more interesting comparison to me with Righi ET al 2020 would be how they did their comparisons between model and observations, and what sensitivity and size range did their data have? Is it the same or different than here?

Page 7, L215: is the RHI data averaged over similar ranges to the cam observations? It would seem this would be required for a reasonable comparison.

Page 7, L225: it's half the scale of the CAM simulations. Also, note that CAM has a wsub minimimum value of a few cm/s, and in upper trop clear sky it's probably at that limit.

One complication is that wsub comes from TKE as you note, which comes from the turbulence scheme. High wsub indicates convection and turbulence would be active, so the pathway for freezing may be very different, as active convection would create liquid that would either be homogenously frozen in the microphysics or frozen with specified size in the macro physics. It would not run through the activation code. I.e. high wsub might just indicate convective outflow. Can you check this?

Page 8, L255: decrease

Page 9, L262: is rhmini set to 80% in the simulations? How is ice formed? With a RH threshold? I am not sure CAM 6 has such a closure. Please state the value. The Gettelman et al 2010 reference is for CAM5.

Page 9, L266: have you shown where in parameter space (RHI, Temp, Di, Ni) the IWC is most biased in the observations? I think it would be great to summarize this in the text. Maybe this comes later?

Page 9, L269: this gets back to sampling. And remember the model represents a distribution, which you should plot I think. There is variability in a single value in the

model. It's not one value.

Page 9, L278: see above. The ice formation mechanism in CAM might not be what you think in the presence of convection at cirrus temps.

Page 9, L281: I do not think using cloud fraction here is wise. Cloud fraction is a function of scale as well as the detection threshold: if the OBS don't see any Di < 65um, that will skew things itself relative to CAM. Maybe you should use something easier to make consistent between model and OBS. I'm not sure what that is, maybe Ni since it is an in cloud only quantity.

Also, can you sample aerosols in cloud in the Obs? If not, then are you filtering that out of the model?

Page 10, L300: it's not clear to me there was much shown with the CAM6 free running simulations. I assume you will summarize Any differences later in this section?

Page 10, L304: since zonal locations in each latitude band are narrow, are you sure this is general and not just a land-sea contrast? Is it anthropogenic aerosols or just land v. ocean?

Page 11, L325: could the model just be producing smaller crystals that are not seen in the observations when Na100 is large?

Page 11, L330: this statement is a bit to grandiose: it's not really comprehensive and there are a limited set of factors.

Page 11, L331: some more summary of what the results actually said here is warranted. What did you discover about geographical locations?

Also, I'm not sure zonal averages are that helpful if the mix regimes, as noted earlier.

Page 11, L332: I'm still not sure the comparisons and elimination from the model of small ice are done correctly.

Page 25, L592: I would recommend collapsing this to put the observations (maybe as a shaded region) on the same plot as the simulations. It makes comparisons easier and results in fewer figure panels.

Page 28, L610: Figures 6 and 7: maybe you could normalize these to get them on the same scale. What is going on in CAM with regular frequency peaks in temperature?
* * *

---

## Referee Comment (RC2) · Anonymous Referee #2 · 1 Sep 2020

Review of "Effects of Thermodynamics, Dynamic and Aerosols on Cirrus Clouds Based on In Situ Observations and NCAR CAM6"

Patnadue et al.

This paper represents a nice study of microphysical observations versus climate modeling. The paper is well written and clear. I shear some of the same concerns as the other reviewer "Andrew Gettelman" but I think this paper could be published after all comments are addressed. Below are my main comments and concerns.

Line 65: Is the reason for increased crystal size with increased aerosols due to the competition between homogeneous and heterogeneous ice nucleation and that you

[Figure]

actually have fewer ice crystal concentration in the polluted environment?

Line 73: How does the increase in aerosol number concentration help increase the size of ice crystals?

Line 86: I suggest to rephrase: "….and found a decrease in Ni with increasing aerosol concentration due to the ……."

Line 131-133: Are there differences between ocean and land as well?

Line 148: If you used the microphysical version including graupel and hail (MG3) you should cite Gettelman et al 2019, "The impact of rimed ice on Global and Regional Climate" in JAMES

Line 154. Do the nudged runs also use prescribed sea-surface temperature?

Line 166: If you disregard the smallest sizes in the observations to define in-cloud conditions, but account for all sizes in the model when defining in-cloud conditions how often do you miss observed in-cloud conditions compared to modeled in-cloud conditions?

Line 167: Why is the additional constraint on cloud fraction not used for the CAM-nudged

Line 181: When mentioning figure 4 I suggest adding a sentence stating that 3 top rows are observations and 3 lower rows are model. Perhaps you can add a label in the figure as well.

Line 187: Mention that the CAM6-nudg data is the 3 bottom rows

Line 207: Did you include the pre-existing ice option by Shi et al in the simulations? Perhaps you should mention that here.

Line 210: I suggest using same color scale between figure 6 and figure 7

Line 211: What is the cause of the systematic "wave" showing up in the tropical RHi in

Figure 7?

Line 212: It is difficult to see the difference between the solid line and the dashed line in figures 6 and 7.

Line 224: Figure 8 and 9 (and other figures with variance of w). Since this value is never negative, I suggest starting the scale at zero.

Line 224. The 200 seconds of data corresponding to 46 km, is that true for all flights?

Line 252: Figure 11: Are the number of samples normalized for the colorbar? I also suggest label the top row as observations, middle as CAM6-nudg and bottom as CAM6-free data.

Line 273: Figure 12. I do not see a large positive correlation between Di and ïĄşw. I would suggest state: ". ....which differs from the slight observed positive. . ..."

Figure 13. Label the two left columns as Na500 and the two right columns as Na100. In the caption, figures q-t are not described.

––––––––––––––––––––––––––––––

---

## Author Comment (AC1) · 8 Nov 2020

**Response to the Reviewers**
Format: The reviewers' comments are quoted in italic
Line number in the response refers to the revised manuscript with tracked changes
Quotation in red color stands for revised/added text in the revised manuscript

**Overall comment:**

We thank the reviewers for their detailed comments. We conducted a substantial revision to the manuscript, including (1) averaging the observations by every 430 seconds (i.e., ~100 km horizontal scale) and comparing them with grid-mean quantities from the simulations (shown in **new Figures 5, 7 to 16**), (2) showing the comparisons of particle size distributions between observations and simulations in **new Figure 4**, and (3) moving the original comparisons between 1-Hz observations and in-cloud quantities of simulations into supplementary material.

Below are our individual responses to the reviewers' comments.

**Response to the Reviewer 1's comments:**

*Review by Andrew Gettelman, NCAR*

*In general, this is a very well written cutting-edge analysis of a comparison between observations of upper tropospheric ice from aircraft and an advanced large-scale global climate model. However, I have some questions on how the analysis was conducted and the sampling of the models and observations. The work is very cutting edge in the detail of the comparisons, but probably needs a few more pieces of description, and more information perhaps to back up the analysis. I think the manuscript needs some substantial revisions before it will be acceptable for publication in Atmospheric Chemistry and Physics.*

**Major comments**

*1) In general, I am concerned that the comparison between the model and observations is not sampling them the same way. It does not seem as if the observations are averaged over a model grid box length before actually reporting values. This should be clarified.*

We thank you for this comment and it is a valid concern. Previously, our comparisons were conducted between 1-Hz observations and the in-cloud quantities from CAM6 output (i.e., variables such as "ICINC" and "ICIMR"). That comparison method was similar to the one used in Righi et al. (2020), which compared 1-Hz aircraft observations with in-cloud quantities from simulations of a global climate model (GCM). Considering the suggestion of the reviewer, we use a new method in the main text to compare observations and simulations on more similar scales. This method compares 430-second averaged observations (i.e., ~100 km horizontal scale based on a mean true air speed of ~230 m/s) and grid-mean quantities from the CAM6 simulations. The new comparison results are shown in **revised Figures 5, 7 to 16**. In addition, we moved the original comparisons to supplementary material as **Figures S4, S7 to S12**.

We described the new method in sections 2.2 (line 186 – 194): "In order to examine observations and simulations on more comparable scales, a running average of 430 seconds was calculated for meteorological parameters (i.e., temperature and RHi) and microphysical properties (i.e., IWC, Ni and Di), which translates to ~100 km horizontal scales since the mean true air speed for all campaigns was 230 m/s (supplementary Figure S2). Grid-mean quantities from model output are used in comparisons with observations, including "IWC", "NUMICE", "QSNOW" and "NSNOW", which are mass and number concentrations of ice particles and snow, respectively. Another type of comparison between 1-Hz observations and in-cloud quantities from model output is shown in the supplementary material. Both methods have been previously used in model evaluation, such as D'Alessandro et al. (2019) which compared 200-s averaged aircraft observations with simulated grid-mean quantities, and Righi et al. (2020) which compared 1-Hz aircraft observations with simulated in-cloud quantities."

(line 209 – 211) "In-cloud conditions in simulations are defined by concurring conditions of IWC $> 10^{-7}$ g m$^{-3}$ and Ni $> 10^{-4}$ L$^{-1}$ based on size-restricted grid-mean quantities. These thresholds are the lower limits from observations after calculating the 430-s averages."

*2) But fundamentally, when this is done, you get a distribution of individual observations which make up a size distribution. In CAM6, it reports a single mass and number for clouds in a grid box. But this represents a size distribution itself. These size distributions can be compared: it may be that the mass and number is distributed differently than the observations. Gettelman et al 2020, in Press, JGR illustrates this method. You can plot the size distributions from the model by reconstructing the distributions from the model equations.*

This is a very helpful comment. We applied the method suggested by the reviewer and reconstructed size distributions for both ice and snow before and after applying the size restriction that excludes ice and snow particles with diameters less than 62.5 µm. We added a **new Figure 4** (below) and more discussions to section 2.2 (line 215 – 220): "To visualize the impact of the size truncation on simulated data, we employed methods similar to Gettelman et al. (2020) and reconstructed the simulated particle size distributions for snow and ice in Figure 4, using gamma functions from Morrison and Gettelman (2008). Note that prior to restricting the diameters of ice and snow particles to ≥ 62.5 µm, the number density for combined ice and snow is overestimated for smaller particles (< 1000 µm) and underestimated for larger particles (≥ 1000 µm). After applying size restriction, the simulated size distribution for combined ice and snow (dashed purple line) becomes more similar to observations due to the reduction of number density of small particles.

[Figure]

**Figure 4.** Observed size distribution (black line) and reconstructed size distributions from simulated ice (blue) and snow (cyan). Both full size range (solid lines) and truncated size range of diameters ≥ 62.5 µm (dashed lines) are shown for simulated hydrometeors. Size distributions for combined ice and snow in the simulations (purple) are also shown before and after the size restriction.

References added:

Morrison, H. and Gettelman, A.: A new two-moment bulk stratiform cloud microphysics scheme in the community atmosphere model, version 3 (CAM3). Part I: Description and numerical tests, *J. Clim.*, 21(15), 3642–3659, doi:10.1175/2008JCLI2105.1, 2008.

Gettelman, A., Bardeen, C. G., McCluskey, C. S., Järvinen, E., Stith, J., Bretherton, C., et al. (2020). Simulating Observations of Southern Ocean Clouds and Implications for Climate. *Journal of Geophysical Research: Atmospheres*, 125, e2020JD032619. https://doi.org/10.1029/2020JD032619

*3) This leads to my next point. The lack of observations below 62 microns diameter means there may be a significant amount of missing ice in the observations. Have you used the distribution functions to truncate the model size, number and mass to reflect this? If not, it's going to make the comparison even worse. If so, there may be an issue just with the size distributions themselves (see earlier point).*

We appreciate you pointing out this issue. To answer your question, the simulated values we used for model evaluation, i.e., IWC, Ni and Di, all reflect the size restriction to ≥ 62.5 µm. To clarify any confusion, we elaborate on our methods for model size constraint in section 2.2 (line 202 – 207): "Simulated ice and snow are restricted to ≥ 62.5 µm based on the size cut-off of the Fast-2DC probe by applying methods from Eidhammer et al. (2014). Based on their equations 1 to 5, we followed their assumption that the shape parameter µ equals 0 when calculating the slope parameter λ. Mass and number concentrations of ice and snow are further calculated based on integrals of incomplete gamma functions from 62.5 µm to infinity. The simulated values of

IWC, Ni and Di are calculated based on the combined ice and snow population after applying the size restriction."

*4) I find it hard to imagine that a climate model can produce 30x less ice than observed and still produce reasonable radiative fluxes, particularly for the Outgoing Longwave Radiation. Maybe I am wrong. It's even worse if the model has small ice that is not seen in the observations, but it's explainable if you have removed all the small crystals already. But since you are actually deriving IWC from a distribution of particles, it seems possible that the assumptions you are making may be very wrong (i.e. the number/size v. IWC relationship). What is the uncertainty here? See specific comment below.*

We appreciate you pointing out this potential issue. After using the new method of comparing 430-s averaged observations and grid-mean quantities from model output, the differences between simulations and observations become smaller compared with the original method, as shown in revised **Figure 5**. We described these comparisons in section 3.1 (line 238 – 251): "The simulations are further compared with averaged observations at a similar horizontal scale of ~100 km. After applying 430-s running averages for observations, the average IWC and Ni values decrease by 0.5 – 1.5 orders of magnitude compared with 1-Hz observations depending on temperature and geographical region. Hemispheric differences are mostly consistent between 1-s and 430-s averaged observations except for polar regions. … The simulated IWC, Ni and Di also show smaller differences between hemispheres and latitudes. The CAM6-nudg data underestimate and overestimate IWC in the NH and SH by 0.5 – 1 orders of magnitude, respectively, with the largest discrepancies in the midlatitudes. The simulations overestimate Ni in the tropics and polar regions in both hemispheres by 0.5 – 1 orders of magnitude, and overestimate Ni in the southern hemispheric midlatitude by 1 – 2 orders of magnitude. The simulated Di is about half of the observed values in most regions except polar regions."

(line 258 – 260) "A sensitivity test is conducted by comparing 1-Hz observations with in-cloud quantities from model output (supplementary Figure S4). Larger differences are seen between simulated and observed IWC and Ni in Figure S4 compared with Figure 5. The directions (i.e., positive or negative) of model biases of IWC, Ni and Di are generally consistent in both comparisons."

*5. Also, there is very little commentary on whether the nudged or free running reproduces observations better. At least not in the conclusions. I think there were some vague comments about the simulations being 'similar'. Is there no appreciable difference?*

Thank you for pointing this out. We added discussions of the two types of simulations in the revised manuscript (line 369 – 379): "Comparing the performance of two types of simulations, both CAM6-nudg and CAM-free show bimodal distributions for IWC – RHi and Ni – RHi correlations, and they both show positive correlations for IWC – $\sigma_w$ and Ni – $\sigma_w$. This result indicates that the general trends in these correlations are statistically robust and less affected by sampling sizes and geographical locations. For correlations with RHi, the maximum IWC value in CAM6-nudg and CAM-free is lower than the 430-s averaged observations by a factor of 25 and 100, respectively. The maximum Ni value in CAM6-nudg is similar to the 430-s averaged observations, while that value in CAM-free is lower by a factor of 3. For correlations with $\sigma_w$,

there are no significant differences for the maximum IWC between the two simulation types. The maximum Ni value in CAM6-nudg and CAM-free is higher than the 430-s averaged observations by a factor of 3 and 10, respectively. These results show that CAM6-nudg data, which are collocated with flight tracks, produce IWC and Ni values closer to the 430-s averaged observations than CAM-free, possibly due to the variabilities of IWC and Ni in different geographical locations as shown in Figure 5."

We also addressed this topic in section 5 (line 430 – 432): "Both simulation types show similar correlation trends of ice microphysical properties with respect to RHi and $\sigma_w$. CAM6-nudg performs better for representing IWC and Ni magnitudes than CAM-free, possibly due to better collocation between CAM6-nudg and observations."

**Minor (specific) comments**

*Page 2, L43: there are IWC observations and compilations that go back decades. Much of it by Heymsfield. Please cite some earlier work.*

We addressed this comment by rewording the sentence and adding additional references (line 45 – 47): "In situ observations of tropical, midlatitude, and polar cirrus clouds have shown that IWC can vary orders of magnitude depending on the geographical locations (Heymsfield, 1977; Heymsfield et al., 2005, 2017; Mcfarquhar and Heymsfield, 1997; Schiller et al., 2008)."

References added:

Heymsfield, A. J.: Precipitation Development in Stratiform Ice Clouds: A Microphysical and Dynamical Study, J. Atmos. Sci., 367–381, 1977.

Heymsfield, A. J., Winker, D. and van Zadelhoff, G. J.: Extinction-ice water content-effective radius algorithms for CALIPSO, Geophys. Res. Lett., 32(10), 1–4, doi:10.1029/2005GL022742, 2005.

Mcfarquhar, G. M. and Heymsfield, A. J.: Parameterization of tropical cirrus ice crystal size distributions and implications for radiative transfer: Results from CEPEX, J. Atmos. Sci., 54(17), 2187–2200, doi:10.1175/1520-0469(1997)054<2187:POTCIC>2.0.CO;2, 1997

*Page 2, L51: Fu and collaborators and Mitchell and collaborators have done some work with CALIPSO that might be relevant here, particularly for occurrence and particle size. IWC from CALIPSO is harder.*

Thank you for this recommendation. We added several references (line 56 – 62): "Using satellite observations from the Cloud-Aerosol Lidar and Infrared Pathfinder Satellite Observation (CALIPSO), Mitchell et al. (2018) showed the dependence of ice particle effective diameter on temperature, latitude, season and topography. Thorsen et al. (2013) also used CALIPSO data to examine cloud fraction of tropical cirrus clouds and showed dependence on altitude and diurnal cycle. Tseng and Fu (2017) used CALIPSO and Constellation Observing System for Meteorology, Ionosphere, and Climate (COSMIC) data and found that the tropical cold point

tropopause temperature is a controlling factor of cirrus cloud fraction in the tropical tropopause layer."

References added:

Mitchell, D. L., Garnier, A., Pelon, J. and Erfani, E.: CALIPSO (IIR-CALIOP) retrievals of cirrus cloud ice-particle concentrations, Atmos. Chem. Phys., 18(23), 17325–17354, doi:10.5194/acp-18-17325-2018, 2018.

Thorsen, T. J., Fu, Q., Comstock, J. M., Sivaraman, C., Vaughan, M. A., Winker, D. M., and Turner, D. D.: Macrophysical properties of tropical cirrus clouds from the CALIPSO satellite and from ground-based micropulse and Raman lidars, J. Geophys. Res. Atmos., 118, 9209–9220, doi:10.1002/jgrd.50691, 2013.

Tseng, H.-H. and Fu, Q.: Temperature control of the variability of tropical tropopause layer cirrus clouds. Journal of Geophysical Research: Atmospheres, 122, 11,062–11,075. https://doi.org/10.1002/2017JD027093, 2017.

*Page 4, L107: CAM6-nudg? Typo? Seems like you are using this throughout, but I don't see it defined.*

We added clarification (line 119 – 121): "Maps comparing the flight tracks of in situ observations and the collocated CAM6 nudged simulations (hereafter named "CAM6-nudg" data) are shown in Figure 1."

*Page 4, L110: Are you going to test the impact of this assumption? Does CAM6 have a similar or close relationship between IWC and number/size?*

For the purposes of this paper, we did not test the impact of this assumption. We added a sentence to make this clear (line 441 – 442): "The assumption of ice mass and dimension relationship from Brown and Francis (1995) may also lead to uncertainties due to various ice habits."

CAM6 does not use such relationship between IWC and size. Predicted IWC and $N_i$ are based on a two-moment microphysics scheme (Gettelman and Morrison, 2015), thus size is predicted but not based on the same assumption as the observations.

*Page 4, L113: since CAM6 has 2 moments and a distribution, wouldn't it be wise to show the size distribution from observations (not just the mean diameter) and from CAM to see if there are biases in the shape or in parts of the size distribution?*

Thank you for this comment. As mentioned above, we added **Figure 4** that addresses this concern by comparing the observed size distribution with the reconstructed model size distribution both before and after applying the size constraint.

*Page 4, L119: 1hz is only maybe 200m horizontal. How can that be compared with a global model at 100km resolution?*

As mentioned above, we now use a new method that compares 430-s averaged observations with grid-mean quantities from simulations.

*Page 4, L121: does CAM6 also use Murphy and Koop? Could that be an issue (probably not).*

This is a good question, and one we should have been more clear addressing. It was noted in the original Figure 7 caption that we used the equations from Murphy and Koop (2005) for calculation of RHi in both observations and simulations. We added a clarification in section 2.2 (line 196 – 197): "Simulated RHi is calculated from simulated specific humidity and temperature, and the calculation of saturation vapor pressure with respect ice is based on the equation from Murphy and Koop (2005)."

*Page 4, L123: whereas ă˘Aˇ˘T> where*

Revised. (line 137 – 138) "Measurements are separated by cloud condition where in-cloud condition is defined by the presence of at least one ice crystal from the Fast 2-DC probe ($N_i > 0$ $L^{-1}$)."

*Page 4, L128: you might note that some latitudes have some very different regimes.*

We addressed this by adding a sentence in section 2.1 (line 142 – 143): "The majority of observations in the SH midlatitude and tropical regions are located over the oceans, while the observations of NH midlatitude and polar regions are predominantly over land."

*Page 5, L138: I'm not sure I found where this size cutoff is noted. Is it wise to proceed with only half the size distribution? Seems like that would strongly affect how forceful you can make the conclusions. It also means the model needs to be sampled carefully.*

Thank you for pointing this out. The original manuscript did mention the size cutoff in a paragraph before that sentence. This description of size restriction is now in line 127 – 128: "In order to mitigate the shattering effect, particles with diameters $< 62.5$ μm (i.e., first two bins) are excluded in the Fast-2DC measurements when calculating IWC, $N_i$ and $D_i$."

*Page 5, L140: So, I assume this is then a sampling bias to your observational data set?*

Yes, we added a sentence to address this (line 153 – 155): "The higher $D_i$ in this study also leads to lower range of $N_i$ ($0.01 – 1000$ $L^{-1}$) and higher range of IWC ($10^{-5} – 10$ g m$^{-3}$) compared with that previous study (i.e., $N_i$ from $0.1 – 10^5$ $L^{-1}$ and IWC from $10^{-7} – 1$ g m$^{-3}$), representing the sampling bias towards larger particles in this study."

*Page 5, L150: it would be also worth noting the most ice relevant adjustments in CAM6: the use of Hoose et al mixed phase ice nucleation, and Shi et al. modifications for pre-existing ice.*

To address this comment, we added a sentence in section 2.2 (line 168 – 171): "The model uses Wang et al. (2014b) for ice nucleation, which implemented and improved Hoose et al. (2010) by considering the probability density function (PDF) of contact angles for the classical nucleation theory. The model also uses Shi et al. (2015) for modifications of pre-existing ice."

References added:

Hoose, C., Kristjánsson, J. E., Chen, J. P. and Hazra, A.: A classical-theory-based parameterization of heterogeneous ice nucleation by mineral dust, soot, and biological particles in a global climate model, J. Atmos. Sci., 67(8), 2483–2503, doi:10.1175/2010JAS3425.1, 2010.

Wang, Y., Liu, X., Hoose, C., and Wang, B.: Different contact angle distributions for heterogeneous ice nucleation in the Community Atmospheric Model version 5, Atmos. Chem. Phys., 14, 10411–10430, https://doi.org/10.5194/acp-14-10411-2014, 2014b.

*Page 6, L169: this is a pretty substantial limitation and should be noted much earlier in the text and even have caveats in the abstract. Showing model size distributions, I think is critical here. Are you calculating simulated IWC without the small particles? Would that skew the results?*

We appreciate your comment, and we agree it is a limitation of this study. To address this, we added a comment on this limitation in our abstract (line 14 – 15): "Observed and simulated ice mass and number concentrations are constrained to $\geq$ 62.5 µm to reduce potential uncertainty from shattered ice in data collection."

As mentioned above, we added clarification of the size restriction to model output in section 2.2. A new Figure 4 is added to illustrate the particle size distributions before and after size restriction. We also added a comment on this caveat in section 4 (line 439 – 444): "It is possible that small ice crystals < 62.5 µm may have formed under high Na but are excluded due to the size constraint. … These caveats call for more investigation on small ice measurements, INP measurements at temperature $\leq$ -40°C, and measurements of various ice habits."

*Page 6, L172: as noted above I think the method here is critical. Please describe it. I would feel more comfortable if you show he size distribution by deriving it from the mu and lambda of the gamma distributions, to understand the truncation issue.*

This comment has been addressed above by revising section 2.2 and adding the new Figure 4.

*Page 6, L177: I suggest that this is a major limitation of correlating Na500 with INP at the temperatures you are working with: INP activation is a strong function of temperature.*

This is a valid concern. We added a sentence to the discussion section to address this as a caveat (line 440 – 441): "Additionally, because INP activation is highly dependent upon temperature, we acknowledge the limitation of using $Na_{500}$ to indicate INP concentrations."

*Page 6, L187: do you want to comment why here? SH is oceanic, NH is continental.*

This is a helpful comment. We addressed this by adding this sentence (line 234 – 236): "These hemispheric differences in midlatitudes may be due to airmass differences between NH (more continental) and SH (more oceanic) and/or more anthropogenic emissions in the NH."

*Page 6, L190: tropical regions with colder temps might have more small crystals. Would this affect the CAM v. OBS results? I'm concerned you have not filtered the cam results by truncating size distributions.*

As noted in previous comments, after averaging observations by every 430 seconds, simulated IWC in new **Figure 5** becomes more comparable to the observations compared with our original method that compares 1-Hz observations and simulated in-cloud quantities. In new Figure 5, the tropics actually show smaller differences between simulated and observed IWC compared with those in the midlatitudes, which suggests that size restriction is not the main reason for model biases in this comparison of IWC.

*Page 6, L194: I find the fact that you can get the OLR right with 10x less ice a little bit strange. Does ice mass not mater as all? Or does the di change compensate? Or the underestimation a product of the Observations really missing a lot of small ice? These don't seem to work the same way. So, I am concerned that you are comparing 1hz IWC v. 100km IWC and this is producing anomalous results.*

As we mentioned above, the model biases of IWC become smaller when comparing grid-mean simulated quantities with 430-s averaged observations. We revised the text for new Figure 5 (line 247 – 251): "The CAM6-nudg data underestimate and overestimate IWC in the NH and SH by 0.5 – 1 orders of magnitude, respectively, with the largest discrepancies in the midlatitudes. The simulations overestimate Ni in the tropics and polar regions in both hemispheres by 0.5 – 1 orders of magnitude, and overestimate Ni in the southern hemispheric midlatitude by 1 – 2 orders of magnitude. The simulated Di is about half of the observed values in most regions except polar regions."

The abstract is also revised (line 16 – 18): "Comparing with averaged observations at ~100 km horizontal scale, simulations are found to underestimate (overestimate) IWC by a factor of 3–10 in the Northern (Southern) Hemisphere."

*Page 7, L208: the more interesting comparison to me with Righi ET al 2020 would be how they did their comparisons between model and observations, and what sensitivity and size range did their data have? Is it the same or different than here?*

Righi et al. (2020) did a comparison between 1-Hz observations and in-cloud quantities from model output. Their analysis is similar to our Figure 6 (RHi versus temperature plot), which showed the average values of RHi, Ni and Di in each temperature bin. One major difference between their study and ours is the size range of ice as they used 3 – 1280 μm. We added clarification on their method in several places.

(line 192 – 194) "Both methods have been previously used in model evaluation, such as D'Alessandro et al. (2019) which compared 200-s averaged aircraft observations with simulated

grid-mean quantities, and Righi et al. (2020) which compared 1-Hz aircraft observations with simulated in-cloud quantities."

(lines 261 – 266) "A previous study by Righi et al. (2020) evaluated the ice microphysical properties in EMAC-MADE3 aerosol–climate model (i.e., ECHAM/MESSy Atmospheric Chemistry-Modal Aerosol Dynamics model for Europe adapted for global applications, 3rd generation) by comparing in-cloud quantities from model output with 1-Hz in situ observations of multiple aircraft field campaigns from 75ºN to 25ºS (Krämer et al., 2009, 2016, 2020). Although that study included more smaller ice particles (3 – 1280 µm) compared with this study, they still showed low biases of simulated Di at 190 – 243 K, low biases of simulated IWC at 205 – 235 K, as well as high biases of simulated Ni above 225 K, ..."

*Page 7, L215: is the RHI data averaged over similar ranges to the cam observations? It would seem this would be required for a reasonable comparison.*

Previously, the comparison was between RHi from 1-Hz observations and CAM6. In the revised manuscript, we averaged the observations by every 430 seconds, including cloud properties and meteorological conditions (such as temperature, RHi and vertical velocity). The **new Figure 7** shows 430-s averaged RHi observations. The clarification is added in section 2.2 (line 186 – 189): "In order to examine observations and simulations on more comparable scales, a running average of 430 seconds was calculated for meteorological parameters (i.e., temperature and RHi) and microphysical properties (i.e., IWC, Ni and Di), which translates to ~100 km horizontal scales since the mean true air speed below -40°C for all campaigns was 230 m/s (supplementary Figure S2)."

*Page 7, L225: it's half the scale of the CAM simulations. Also, note that CAM has a wsub minimimum value of a few cm/s, and in upper trop clear sky it's probably at that limit. One complication is that wsub comes from TKE as you note, which comes from the turbulence scheme. High wsub indicates convection and turbulence would be active, so the pathway for freezing may be very different, as active convection would create liquid that would either be homogenously frozen in the microphysics or frozen with specified size in the macro physics. It would not run through the activation code. I.e. high wsub might just indicate convective outflow. Can you check this?*

This is a great comment. We added a new supplementary **Figure S5** to show the locations where model output wsub exceeds 0.5 m/s, as well as where observed vertical velocity is greater than 1 m/s for in-cloud conditions. Based on this figure, the majority of in-cloud samples do not show very high wsub or vertical velocity. We pointed out that more future work is needed to track the origin of cirrus clouds formed in both model and observations to distinguish the impacts from convection. We added this discussion in section 3.2 (line 299 – 305): "We further examine the potential impact of convection in simulations and observations. Supplementary Figure S5 shows the locations where w > 1 m/s is seen in the observations as well as where wsub > 0.5 m/s is seen in the CAM6-nudg data for in-cloud conditions. Since wsub in CAM6 is based on the turbulent scheme, higher wsub values indicate that the convection scheme may be active and produce detrained ice in convective outflows. The majority of observed and simulated in-cloud samples do not appear to have high w or wsub, indicating that detrained ice from the convection is

unlikely a significant contribution. More future investigation is needed to track cirrus cloud origins and quantify impacts from convection."

[Figure]

**Figure S5.** (a) Locations of simulated in-cloud samples with wsub > 0.5 m/s, color coded by $\sigma_w$. (b) Locations of observed in-cloud samples with vertical velocity > 1 m/s, color coded by $\sigma_w$ calculated for every 430 seconds.

*Page 8, L255: decrease*

Changed "decreases" to "decrease".

*Page 9, L262: is rhmini set to 80% in the simulations? How is ice formed? With a RH threshold? I am not sure CAM6 has such a closure. Please state the value. The Gettelman et al 2010 reference is for CAM5.*

This is a good question. For the ice cloud fraction parameterization, the $RH_{min}$ parameter was set to 80% and $RH_{max}$ parameter was set to 100% in the CAM6 simulation, which are the same as CAM5. For cirrus clouds, heterogeneous nucleation can begin after reaching a minimum threshold 100%, which technically is 120% when accounting for a sub-grid variability scaling factor of 1.2 (Wang et al., 2014a). The homogeneous nucleation RHi threshold is around 150% –

160%. Beside the RHi thresholds, ice nucleation is also dependent upon additional constraints of temperature and vertical velocity (Liu et al., 2007; Liu and Penner, 2005).

We have revised our discussion of the effects of RHi on microphysical properties (line 341 – 346): "In contrast to observations, both CAM6-nudg and CAM6-free simulations show bimodal distributions of IWC and Ni with the primary peak at 100% RHi and the secondary peak at 80% RHi. The secondary peak at RHi 80% is likely produced by the $RHi_{min}$ parameter reflecting sub-grid scale RHi variance as mentioned above (Gettelman et al., 2010), which was set at the default value (80% RHi) for both simulations. The primary peak at 100% RHi is likely a result of the minimum threshold for heterogeneous ice nucleation being set at 120% as well as a sub-grid variability scaling factor 1.2 being considered (Wang et al., 2014a)."

*Page 9, L266: have you shown where in parameter space (RHI, Temp, Di, Ni) the IWC is most biased in the observations? I think it would be great to summarize this in the text. Maybe this comes later?*

We added **supplementary Figure S1** and a discussion in section 2.1 (line 155– 158): "The relationships of IWC with respect to meteorological conditions (i.e., temperature and RHi) and other microphysical properties (i.e., Ni and Di) are shown in supplementary Figure S1. The distributions of IWC samples are relatively uniform at various temperature and RHi, while more IWC samples are correlated with Di between 100 and 300 µm."

[Figure]

**Figure S1.** Number of samples of log-scale IWC values in the observations related to various meteorological conditions (i.e., (a) temperature and (b) RHi) and microphysical properties (i.e., (c) log-scale Ni and (d) Di). Average IWC values and standard deviations are represented by black lines and whiskers, respectively. Note that all samples are for temperature ≤ -40ºC.

*Page 9, L269: this gets back to sampling. And remember the model represents a distribution, which you should plot, I think. There is variability in a single value in the model. It's not one value.*

We added the particle size distribution in new Figure 4.

*Page 9, L278: see above. The ice formation mechanism in CAM might not be what you think in the presence of convection at cirrus temps.*

We agree and added supplementary Figure S5 as discussed above.

*Page 9, L281: I do not think using cloud fraction here is wise. Cloud fraction is a function of scale as well as the detection threshold: if the OBS don't see any Di < 65um, that will skew things itself relative to CAM. Maybe you should use something easier to make consistent between model and OBS. I'm not sure what that is, maybe Ni since it is an in cloud only quantity. Also, can you sample aerosols in cloud in the Obs? If not, then are you filtering that out of the model?*

We would like to clarify that the cloud fraction plotted in the original Figure 13 and the new Figures 15 and 16 is not the "Cloud Fraction" variable from the model output. The cloud fraction for simulation in these figures is calculated using the number of in-cloud conditions defined by the concurring IWC and Ni values greater than certain thresholds, and normalized by the total number of samples in each temperature – Na bin. Therefore, the definition of cloud fraction in our study actually follows the reviewer's suggestion of using Ni since it is an in-cloud only quantity. This is clarified in section 4.3 (line 382 – 384): "Cloud fraction is calculated in each temperature – Na bin by normalizing the number of in-cloud samples with the total number of samples in that bin for both observations and simulations." The in-cloud condition for simulation is defined here (line 209 – 211): "In-cloud conditions in simulations are defined by concurring conditions of IWC > $10^{-7}$ g m$^{-3}$ and Ni > $10^{-4}$ L$^{-1}$ based on size-restricted grid-mean quantities. These thresholds are the lower limits from observations after calculating the 430-s averages." And to answer your question of aerosol sampling in clouds, yes, the aerosol measurements are available for in-cloud conditions.

*Page 10, L300: it's not clear to me there was much shown with the CAM6 free running simulations. I assume you will summarize Any differences later in this section?*

See our previous responses regarding differences between CAM6-nudg and CAM6-free.

*Page 10, L304: since zonal locations in each latitude band are narrow, are you sure this is general and not just a land-sea contrast? Is it anthropogenic aerosols or just land v. ocean?*

We agree that both land-sea contrast and anthropogenic aerosols are possible reasons for the hemispheric differences. We acknowledge the possibility of both reasons in our revised text (line 410 – 412): "The hemispheric differences between NH and SH midlatitudes indicate a possible role of anthropogenic aerosols and/or land-sea contrast in controlling ice microphysical properties."

*Page 11, L325: could the model just be producing smaller crystals that are not seen in the observations when Na100 is large?*

This is a good question. It is possible that small ice crystals may have formed at high $Na_{500}$ and $Na_{100}$ and have been cut out due to size restriction. We added a clarifying sentence to address this (line 439 – 440): "It is possible that small ice crystals < 62.5 µm may have formed under high Na but are excluded due to the size constraint."

*Page 11, L330: this statement is a bit to grandiose: it's not really comprehensive and there are a limited set of factors.*

We deleted the word "comprehensive" in that sentence.

*Page 11, L331: some more summary of what the results actually said here is warranted. What did you discover about geographical locations? Also, I'm not sure zonal averages are that helpful if the mix regimes, as noted earlier.*

We addressed this comment by adding more summary on regional variations in section 5 (line 450 – 454): "For both observations and simulations, higher ice supersaturations and stronger vertical motions are shown in tropical and midlatitude regions, which possibly lead to increased homogeneous nucleation and convection-generated cirrus, consistent with higher IWC and Ni and lower Di in these regions compared with polar regions. In addition, underestimating aerosol indirect effects in the simulations likely contributes to the underestimation of IWC in the NH."

*Page 11, L332: I'm still not sure the comparisons and elimination from the model of small ice are done correctly.*

See previous comments on model size truncation to remove ice particles smaller than 62.5 µm.

*Page 25, L592: I would recommend collapsing this to put the observations (maybe as a shaded region) on the same plot as the simulations. It makes comparisons easier and results in fewer figure panels.*

We thank you for this recommendation. The original Figure 4 has been moved to supplementary material. The new **Figure 5** now shows both model and observation in each sub-panel for convenience.

*Page 28, L610: Figures 6 and 7: maybe you could normalize these to get them on the same scale. What is going on in CAM with regular frequency peaks in temperature?*

We have adjusted the color scales for **Figures 6 – 10** so that they are the same. To answer your question on the regular frequency peaks in temperature in the simulations (**new Figure 8**), we attribute this discontinuity in temperature to larger distances between model vertical levels in the upper troposphere over the tropics. We added this clarification (line 287 – 288): "Note that the simulation samples in the tropical regions show peak frequencies at certain temperatures due to larger bin sizes of pressure levels in the lower latitudes."

**Response to the Reviewer 2's comments:**

Anonymous Referee #2

*This paper represents a nice study of microphysical observations versus climate modeling. The paper is well written and clear. I shear some of the same concerns as the other reviewer "Andrew Gettelman" but I think this paper could be published after all comments are addressed. Below are my main comments and concerns.*

*Line 65: Is the reason for increased crystal size with increased aerosols due to the competition between homogeneous and heterogeneous ice nucleation and that you actually have fewer ice crystal concentration in the polluted environment?*

Great question. Based on the Chylek et al. (2006) study, they speculate that in the polluted environment, more ice nuclei form ice particles via heterogeneous nucleation, therefore reducing the amount of excess water vapor over ice saturation and making homogeneous nucleation a more difficult pathway. We added this clarification (line 79 – 83): "Chylek et al. (2006) showed an increase in ice crystal size during the more polluted winter months compared with cleaner summer months over the eastern Indian Ocean, which the authors speculate to be due to heterogeneous nucleation occurring at lower ice supersaturation compared with homogeneous nucleation, therefore reducing the ambient ice supersaturation magnitude and making homogeneous nucleation a more difficult pathway."

*Line 73: How does the increase in aerosol number concentration help increase the size of ice crystals?*

This is an excellent question, and one we also asked ourselves. In a closed system with limited water vapor supply, one would expect ice crystal number concentration (Ni) and mean diameter (Di) to be anti-correlated, similar to the conventional Twomey effect. In our analysis of the in-situ observations, cirrus clouds may have experienced more complex ambient conditions. One possible scenario is that cirrus clouds from in-situ observations could have experienced a series of different magnitudes of ice supersaturation in their evolution history. This could have induced several ice nucleation events, forming new ice particles in addition to the preexisting ice particles. As a result, Ni would increase, and Di would also increase due to the growth of the previously formed ice particles. Previous observation-based studies on cirrus cloud evolutionary trend (Diao et al., 2013, 2014) support the increase of both Ni and Di during cirrus evolution. In the study of Diao et al. (2013), we found that as cirrus clouds evolve from nucleation phase to early growth and later growth phases, both Ni and Di increase.

References mentioned above:

Diao, M., Zondlo, M. A., Heymsfield, A. J., Beaton, S. P. and Rogers, D. C.: Evolution of ice crystal regions on the microscale based on in situ observations, Geophys. Res. Lett., 40(13), 3473–3478, doi:10.1002/grl.50665, 2013.

Diao, M., Zondlo, M. A., Heymsfield, A. J. and Beaton, S. P.: Hemispheric comparison of cirrus cloud evolution using in situ measurements in HIAPER Pole-to-Pole Observations, Geophys. Res. Lett., 41(11), 1–8, doi:10.1002/2014GL059873, 2014.

*Line 86: I suggest to rephrase: ": : :.and found a decrease in Ni with increasing aerosol concentration due to the : : :: : :"*

We revised this sentence (line 98 – 100): "Shi et al. (2015) added the effects of pre-existing ice into the Community Atmosphere Model Version 5 (CAM5) and found a decrease in Ni with increasing aerosol concentration due to the reduction of homogeneous nucleation frequency."

*Line 131-133: Are there differences between ocean and land as well?*

This is a good question. Based on the map of flight campaigns (their Figure 1) from Krämer et al. (2020), our dataset has more sampling over the ocean compared with theirs, especially for sampling over the Pacific Ocean from the HIPPO Global campaign.

*Line 148: If you used the microphysical version including graupel and hail (MG3) you should cite Gettelman et al 2019, "The impact of rimed ice on Global and Regional Climate" in JAMES*

For the simulations in this paper, we used the MG2 microphysics scheme. As you noted, this does not include graupel and hail. Therefore we revised that sentence (line 164 – 166): "An improved bulk two-moment cloud microphysics scheme has been implemented (Gettelman and Morrison, 2015) that replaces diagnostic treatment of rain and snow with prognostic treatment of all hydrometeors (i.e., rain and snow)."

*Line 154. Do the nudged runs also use prescribed sea-surface temperature?*

Yes. The details are described in section 2.2 (line 175 – 176): "All simulations are conducted using prescribed sea-surface temperature and present-day aerosol emissions and include a 6-month spin-up time."

*Line 166: If you disregard the smallest sizes in the observations to define in-cloud conditions, but account for all sizes in the model when defining in-cloud conditions how often do you miss observed in-cloud conditions compared to modeled in-cloud conditions?*

This is a good question. We use **supplementary Figure S6** to contrast with Figure 12 for the impact of including small ice particles in the simulations. Figure S6 is similar to Figure 12, except for using ice crystals > 1 µm in the analysis for simulated in-cloud data. Figure S6 shows a 4% increase of number of samples for simulated in-cloud conditions compared with Figure 12 (restricted to ≥ 62.5 µm) (line 318 – 320): "When using a lower size cut-off (1 µm) of ice particles for the simulation data, the number of in-cloud samples increases by 4% (supplementary Figure S6)."

*Line 167: Why is the additional constraint on cloud fraction not used for the CAM-nudged?*

We appreciate your question, and we should have clarified. This constraint was indeed used on both CAM6 simulations. We reworded this sentence to be clearer (line 213 – 214): "An additional constraint on cloud fraction $> 10^{-5}$ was applied to both nudged and free-running simulations to exclude extremely low values."

*Line 181: When mentioning figure 4 I suggest adding a sentence stating that 3 top rows are observations and 3 lower rows are model. Perhaps you can add a label in the figure as well.*

Thank you for this recommendation. Since both reviewers mentioned the difficulty with this figure, the new **Figure 5**, now includes both observations and simulations in each sub-panel.

*Line 187: Mention that the CAM6-nudg data is the 3 bottom rows*

We appreciate your recommendation. As we mentioned above, the new Figure 5 now includes both observations and simulations in each sub-panel.

*Line 207: Did you include the pre-existing ice option by Shi et al in the simulations? Perhaps you should mention that here.*

Yes, the model did include the effects of pre-existing ice from Shi et al. (2015). We added this clarification in the methods section (line 170 – 171): "The model also uses Shi et al. (2015) for modifications of pre-existing ice."

*Line 210: I suggest using same color scale between figure 6 and figure 7*

Figures 6 – 11 are revised to use the same color scale.

*Line 211: What is the cause of the systematic "wave" showing up in the tropical RHi in Figure 7?*

This is a good question. We added an explanation of this feature (line 287 – 288): "Note that the simulation samples in the tropical regions show peak frequencies at certain temperatures due to larger bin sizes of pressure levels in the lower latitudes."

*Line 212: It is difficult to see the difference between the solid line and the dashed line in figures 6 and 7.*

We addressed this issue by making both lines thicker.

*Line 224: Figure 8 and 9 (and other figures with variance of w). Since this value is never negative, I suggest starting the scale at zero.*

Thank you for this recommendation. We changed the ordinate range to start from zero for **Figures 9 to 11**.

*Line 224. The 200 seconds of data corresponding to 46 km, is that true for all flights?*

The mean true air speed of ~230 m/s was calculated based on the entire dataset of seven campaigns for all temperatures ≤ -40°C. We added supplementary **Figure S2** to show the mean true air speed at various temperatures averaged for the entire dataset. For each flight, the mean true air speed in the upper troposphere is around this value.

*Line 252: Figure 11: Are the number of samples normalized for the colorbar? I also suggest label the top row as observations, middle as CAM6-nudg and bottom as CAM6-free data.*

The original figure 11 was not normalized, and the color code shows the exact number of samples in each bin. We added labels for each row in the new **Figures 12 – 14**, to distinguish between different datasets.

*Line 273: Figure 12. I do not see a large positive correlation between Di and w. I would suggest state: ": : :..which differs from the slight observed positive: : :.." Figure 13. Label the two left columns as Na500 and the two right columns as Na100. In the caption, figures q-t are not described.*

Thank you for this recommendation. We revise this sentence (line 363 – 364): "This slight positive $D_i$ - $\sigma_w$ correlation is likely due to the growth of ice particles as cirrus clouds evolve with continuous updrafts that supply excess water vapor above ice saturation, ..."

We separated the original Figure 13 into the new **Figures 15 and 16** for $Na_{500}$ and $Na_{100}$, respectively. We added description to the last row in Figure 15 caption.

---

## Author Response (AR2)

**Response to the Reviewer**

Format: The reviewer's comments are quoted in italic

Line number in the response refers to the revised manuscript with tracked changes

Quotation in red color stands for revised/added text in the revised manuscript

We thank the reviewer Dr. Andrew Gettelman for the additional comments. Below is our response to each of the comments, and the revisions to the main text accordingly.

*I think the authors have done a very good job responding to the reviews. I appreciate that they have averaged the data to be near the same scale now between models and observations, and that truncation has been applied to ice with a size threshold representing the observations. I think this will be publishable with a few further clarifications as noted below.*

*I still think that for cloud mass and number, you need to use in-cloud values. Unless you are also sampling zeros from the observations (i.e. the running sample includes zeros when there is not cloud). Please clarify.*

Yes, we included the zero values of Ni and IWC (i.e., clear-sky condition) when averaging into coarser scales. This is why we are comparing the 100-km scale observations with grid-mean quantities from model output. We included a clarification in line 186 – 187: "When applying the running average, both in-cloud and clear-sky conditions (i.e., where Ni and IWC values are zero) are included in the averages."

*For Figure 4, I don't see how truncation for sizes greater than 62.5 microns will INCREASE or DECREASE the number of ice crystals in bins larger than this. I think there is something incorrect about the truncation.*

This is a very good question. We can see that the original Figure 4 in the previously revised manuscript can be confusing. In that figure, we applied new gamma functions to ice and snow after size truncation and obtained new size distributions based on the size-constrained Ni and IWC. That is why that even for particles larger than 62.5 μm, the number concentrations per bin size (unit: $L^{-1}\ \mu m^{-1}$) of the new gamma functions were different from the original gamma functions. Because that original figure could be misleading, we revised it to a new Figure 4 in order to show the impact of size truncation on the same gamma functions as the reviewer suggested, instead of calculating new gamma functions. In addition, we added two subpanels to show the impact of size truncation on the PDFs of Ni and IWC for all model output. Below is the new Figure 4 and the revised text describing it.

Line 206 – 212: "To visualize the impact of the size truncation on simulated data, we employed methods similar to Gettelman et al. (2020) and reconstructed the simulated particle size distributions for snow and ice in Figure 4 a, using gamma functions from Morrison and Gettelman (2008). Compared with the observations, the number density for combined ice and snow is overestimated for smaller particles (< 400 µm) and underestimated for larger particles (> 1000 µm). After applying size restriction, the PDF of simulated Ni and IWC show increasing probability of small Ni and decreasing probability of small IWC due to the removal of small particles (Figure 4 b and c)."

[Figure]

**Figure 4.** (a) Observed size distribution (black line) and reconstructed size distributions from simulated ice (blue) and snow (cyan). Size truncations to diameters < 62.5 µm (dashed lines) are shown for simulated hydrometeors, while the remaining particles (≥ 62.5 µm) (solid lines) are used for comparisons with observations. Size distributions for combined ice and snow in the simulations (purple) are also shown before and after the size restriction. (b) and (c): PDFs of Ni and IWC in the simulation before and after size truncation.

*The ice is handled better, (truncation), but I still think there is something missing in the comparisons: the factor of 3-10 difference seems difficult to understand at the large scale. I think there is still something apples-to-oranges about the comparisons (see comment above). The comment below provides an avenue to address this.*

*You cannot really say that the models underestimate IWC (e.g., line 395, line 408). They underestimate IWC and number in the size range observed from the observations, but that could be a bias in the size distribution. The conclusions should probably be clarified on this point with making the deviation more specific to a size range.*

We agree with the reviewer that the comparison result is only applicable to the size range being evaluated. We added this clarification in section 5 Discussion and conclusions (line 402 – 404): "Differences in the particle size distribution, such as lower number density of larger particles (> 1000 µm) in the simulation

(Figure 4 a), may also contribute to the underestimation of IWC by the simulation. All the comparison results on IWC, Ni and Di are only applicable to the size range being evaluated ($\geq$ 62.5 µm)."